# Modeling the disruption of respiratory disease clinical trials by non-pharmaceutical COVID-19 interventions

Simon Arsène[1], Claire Couty [1,4], Igor Faddeenkov[1,4], Natacha Go[1,4], Solène Granjeon-Noriot[1,4], Daniel Šmít[1], Riad Kahoul [1], Ben Illigens[1,2], Jean-Pierre Boissel[1], Aude Chevalier[3], Lorenz Lehr[3], Christian Pasquali[3] & Alexander Kulesza [1✉]

Respiratory disease trials are profoundly affected by non-pharmaceutical interventions (NPIs) against COVID-19 because they perturb existing regular patterns of all seasonal viral epidemics. To address trial design with such uncertainty, we developed an epidemiological model of respiratory tract infection (RTI) coupled to a mechanistic description of viral RTI episodes. We explored the impact of reduced viral transmission (mimicking NPIs) using a virtual population and in silico trials for the bacterial lysate OM-85 as prophylaxis for RTI. Ratio-based efficacy metrics are only impacted under strict lockdown whereas absolute benefit already is with intermediate NPIs (eg. mask-wearing). Consequently, despite NPI, trials may meet their relative efficacy endpoints (provided recruitment hurdles can be overcome) but are difficult to assess with respect to clinical relevance. These results advocate to report a variety of metrics for benefit assessment, to use adaptive trial design and adapted statistical analyses. They also question eligibility criteria misaligned with the actual disease burden.

[1] Novadiscovery SA, Lyon, France. [2] Dresden International University, Dresden, Germany. [3] OM Pharma, Meyrin, Switzerland. [4] These authors contributed equally: Claire Couty, Igor Faddeenkov, Natacha Go, Solène Granjeon-Noriot. ✉email: Alexander.Kulesza@novadiscovery.com

The COVID-19 pandemic and consecutive response measures to contain the spread of SARS-CoV-2 in the form of non-pharmaceutical interventions (NPIs) have changed not only people's life and health[1] but also the process of developing vaccines and potential treatments[2]. This has led to a rapid pursuit of different immunization strategies against the virus[3,4], a surge of drug repurposing and the screening of new treatment candidates[5,6].

Clinical development in non-COVID-19 disease areas, however, has been substantially impaired[7]. Due to the high number of COVID-19 cases during the pandemic in 2020, trial initiation dropped by up to 30% in the USA[8]. During the first wave of the pandemic, more than 1000 trials were stopped as a consequence[9]. Social distancing and quarantine measures have negatively affected patients' participation in clinical trials. The surge in hospitalizations of COVID-19 patients also affected personnel's capacity to conduct trials[10] and has led to incomplete or delayed data collection in ongoing trials with foreseeable difficulties for patient enrollment and follow-up in upcoming trials. Trialists expect that the collateral impact of COVID-19 on clinical trials will persist for several years[11], given that intermittent containment measures are possible beyond the year 2025[12].

This is especially critical for trials investigating diseases of the pulmonary system. About 10% of all trials conducted in Europe in pre-COVID-19 times were on respiratory diseases[13,14]. Due to COVID-19 containment measures that intend to attenuate SARS-CoV-2 transmission, respiratory disease transmission is altered at the population scale, and/or there might be under-reporting of respiratory diseases to healthcare services (see a recent systematic review by Alqahtani et al.[15]). Recent reports show that seasonal dynamics of common respiratory tract infections (RTIs) have almost vanished during the COVID-19 pandemic[16–19]. In England, overall fewer cases of common cold, flu, and bronchitis have been reported during the lockdown[20]. Detection bias due to reduced testing is not a significant confounder as several sources worldwide[21–23] reported a sharp decline in the number of RTIs relative to the number of tests. Hospitalization for acute bronchiolitis in children <1 year old saw a significant reduction, on the order of 70–90%, comparing 2020 with earlier years[24]. For chronic obstructive pulmonary disease (COPD, often triggered by viral infections), healthcare professionals in Europe have reported fewer cases in community and acute hospital settings[25,26], and a decline in asthma exacerbations has been reported as well[27]. While this decline may be regarded as a positive side-effect of the pandemic, it is only temporary, and rebounding of the respiratory disease burden can happen[28,29]. At the same time, respiratory disease prophylaxis and trials across the world are strongly affected by these drastic changes because the design of clinical trials is usually conceived from pre-pandemic settings, e.g., the sample size calculation and the choice of endpoints and eligibility are based on historical interventional and observational studies and do not mirror the current pandemic situation. Therefore, clinical trial feasibility in respiratory diseases remains an open question in the medium term.

Modeling and simulation might be an approach to address the lack in representativity of historical data if forecasts of disease transmission can be joined with clinical trial simulation. For example, simulated clinical trials have provided the means to test a multitude of design choices[30–33] and became a field gaining attraction throughout regulatory agencies[34]. The COVID-19

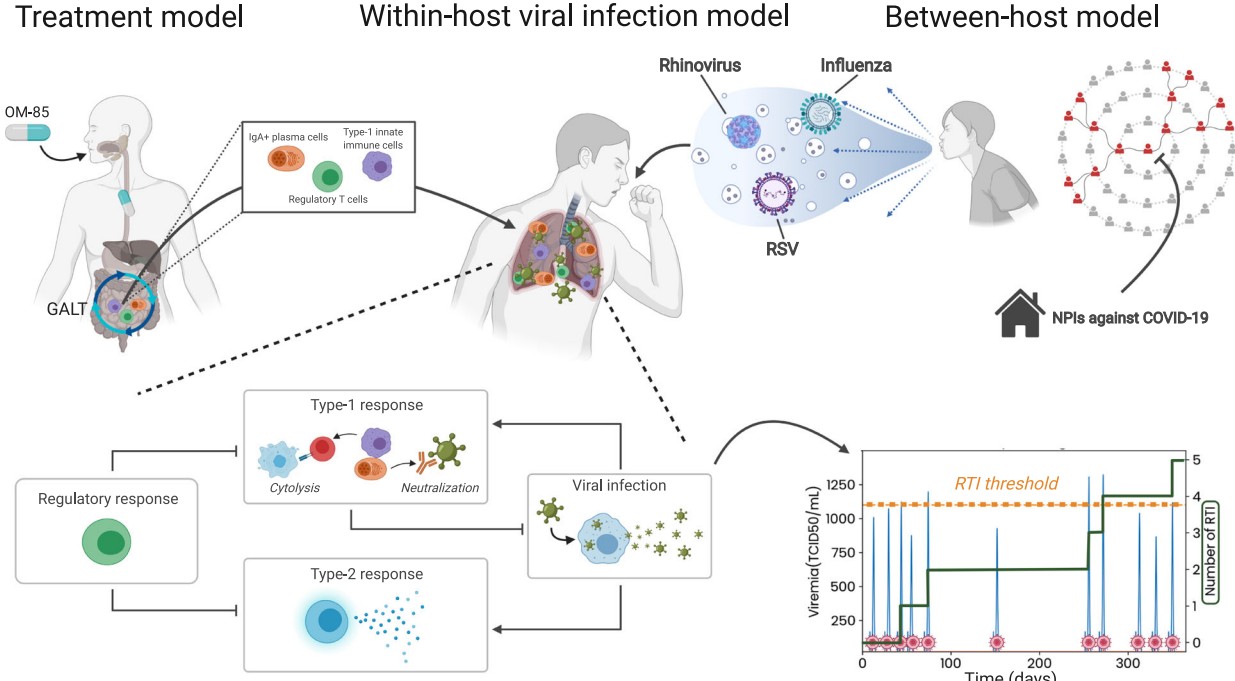

**Fig. 1 Multi-scale in silico approach to incorporate within-host and between-host respiratory tract infection (RTI) model as well as a treatment model with bacterial lysate OM-85. The model is used to assess feasibility of clinical trials in prophylaxis of RTIs during COVID-19 pandemic.** The transmission of the major respiratory pathogens respiratory syncytial virus (RSV), rhinovirus, and influenza type A and B viruses is given by a seasonal susceptible, infected, recovered, and again susceptible (SIRS) model (between-host model). This model is interfaced to a within-host immunology model via a time-dependent instantaneous prevalence of infection triggering or not viral exposure at randomly chosen time points. Individual patients are identified by their age and an immuno-competence meta-parameter impacting the immune response from which infections are included or omitted from the cumulative number of infections depending on viremia. To prevent RTIs, virtual patients are treated with the bacterial lysate OM-85, which acts through a pro-type I immunomodulation mechanism of action and which is described by a physiologically based pharmacokinetics (PBPK) and pharmacodynamics (PD) treatment model with downstream effects in the immunological model. The impact of COVID-19 associated non-pharmaceutical interventions (NPIs) are simulated by scaling of the transmission term in the between-host part of the model. Figure created with BioRender.com.

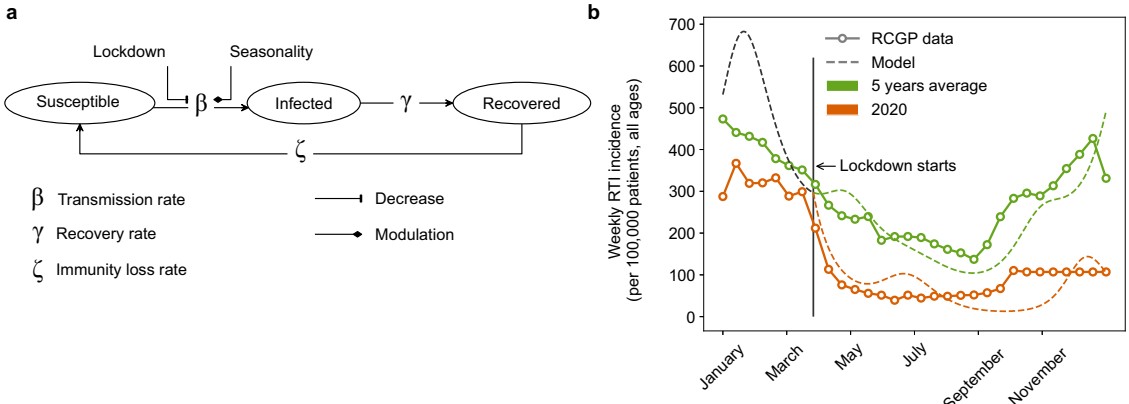

**Fig. 2 Between-host model based on susceptible, infected, recovered, and again susceptible (SIRS) framework allows to reproduce respiratory tract infection (RTI) incidence during non-pharmaceutical interventions (NPIs) to mitigate COVID-19 pandemic. a** Schematic of implemented SIRS model where NPI can be modeled by a decrease of the transmission rate. **b** Comparison of model predictions (dashed lines) and data (solid lines) from Royal College of General Practitioners (RCGP)[41] for RTI weekly incidence (per 100,000 all ages) for the 5 years average (green) and 2020 (orange). Lockdown was started on the 23th of March 2020 in the UK. This date was used to implement the lockdown in the simulations with a decrease of 17.5% of the transmission rate.

pandemic has already transformed the modeling and simulation community. For example, governments rely on mathematical—often epidemiological between-host viral transmission—models to predict the evolution of the pandemic and to take evidence-based decisions[35]. On the other hand, viral kinetic modeling, focusing on the patient immunology and viral infection resolution, can be used to accelerate drug development[36]. As mentioned in a recent review by Karr et al.[37], multi-scale within-host modeling is common, but there are much fewer models that interface within-host models with between-host models because the within-host granularity risks getting lost when integrating to a higher scale. In particular, to our knowledge, there is currently no available modeling approach that can simulate RTI prophylaxis trials under COVID-19 pandemic conditions and, that could serve to better inform respiratory disease trial design and clinical development decisions. Based on the applicability of viral kinetic models on the population and individual scale (i.e., immunology) for a broad variety of viruses, we hypothesized that a mechanistic model as schematized in Fig. 1 could be used for in silico RTI prophylaxis trial simulation and to forecast trial feasibility. After matching known viral disease burden seasonality, intra- and inter-patient variability in RTI resolution and efficacy data, we built a mechanistic model and simulated placebo-controlled in silico trials in 1–5-year-old pediatric patients with recurrent RTIs (RRTI) treated with an immunomodulating bacterial lysate under four different hypotheses of NPI intensities and assessed efficacy and benefit metrics as a function of NPI intensity. We chose the example of OM-85, which is a well characterized (Yin et al.[38]) member of a series of bacterial lysates containing medicinal products for respiratory conditions that have been used in over 120 million patients but need to soon provide new clinical efficacy data in view of an EU referral procedure[39] and explored clinical interpretation, power, sample size and recruitment considerations as aspects of trial feasibility. From the simulations, we conclude on COVID-19 pandemic-related risk mitigation strategies for confirmatory trials concerning this entire class of products and RTI prophylaxis trials in general.

## Results
**Effect of NPIs on RTI disease burden**. Our epidemiological model is based on a compartmental approach describing

susceptible, infected, recovered, and again susceptible (SIRS) individuals and explicitly describes transmission, recovery, and immunity loss rates (Fig. 2a, Methods). We calibrated this model in a parallel manner for the main respiratory viruses (Supplementary Fig. 4, note that the vast majority of RTI are considered to be of viral origin[40]). As representative comparator and validation of the prior virus-specific infection dynamics calibration, we used the 5-year average and the 2019–2020 upper and lower RTI (URTI and LRTI) incidence from the communicable and respiratory disease report 2019–2020 published in the UK by the Royal College of General Practitioners (RCGP)[41] (points and full lines in Fig. 2b). To model NPI, starting at week 12 in 2020, we decreased the scaling factor of the viral transmission rate ($b_0$, Supplementary Methods: Between-host SIRS model) by 17.5% to reproduce the difference between the unperturbed 5-year average and the perturbed 2019–2020 URTI and LRTI incidence with the lockdown. Results of the simulations are displayed as dashed lines in Fig. 2b. Simulations and data show a similar strong decline of the disease incidence with the beginning of the lockdown in the UK during March 2020 (week 10–14)[42] while the 2019–2020 disease burden closely follows the 5-year average (as reported in Iacobucci et al.[20]). With the adjusted transmission rate and otherwise unchanged parameters, the root mean square deviation (RMSD) for the weekly incidence per 100,000 of the simulation vs. data are 82 and 96 (unperturbed simulation vs. 5-year average data and perturbed simulation vs. 2019–2020, data, respectively), which is smaller than the variability within the observed data before lockdown (RMSD of 102 for the 5-year average vs. 2019–2020 data for the time points considered). Furthermore, reproduction of RTI incidence broken down into URTIs and LRTIs (Supplementary Fig. 1) shows convincing capability to describe the effect of transmission perturbation on RTIs. Supported by this agreement, we applied this epidemiological model to modulate the instantaneous hazard of exposure to RTI-causing viruses in our in silico trials with four different NPI scenarios.

**Effect of NPIs on efficacy of RTI prophylaxis**. To represent the effect of different NPI scenarios for a 2-year clinical trial (where the first year is the selection period and the second year is the intervention period), we defined scenarios where the transmission rate is decreased by 5%, 15%, and 25% during the second year (Fig. 3a). As a result, the selection year is unaffected by NPIs, and

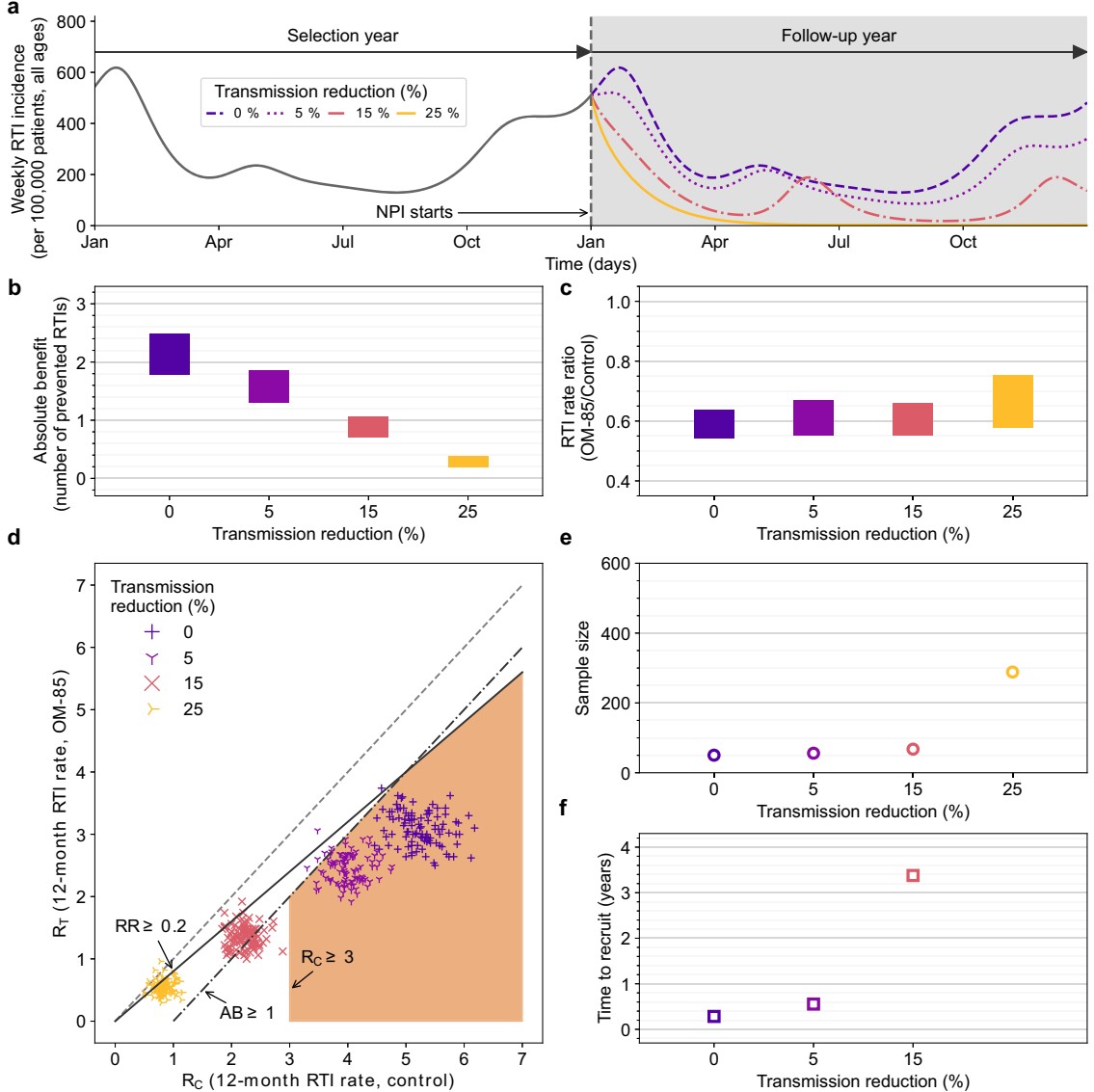

**Fig. 3 Results of in silico clinical trials in prophylaxis of respiratory tract infections (RTIs) with four scenarios of non-pharmaceutical interventions (NPIs) against COVID-19 pandemic with increasing strength (absent - dark purple, mild - light purple, medium - light red, and strong - yellow) modeled by a decrease of the transmission rate parameter (no reduction, −5%, −15% and −25%, respectively).** For all scenarios, the simulations are run for 2 years. Year 1 is the selection year during which patients are screened and possibly included in an in silico trial. There is no NPI during year 1. The NPIs are started at the beginning of year 2 as well as the treatment (ten daily administrations of 3.5 mg of OM-85 from the beginning of the month for 3 consecutive months). RTIs are counted for the complete duration of year 2. **a** Weekly incidence of RTIs per 100,000 is plotted for 2 years of simulations for the four NPI scenarios. **b** Distribution (interquartile range, IQR) of absolute benefit (number of prevented RTIs in year 2) is plotted for the four NPI scenarios. Absolute benefit can be interpreted as the number of prevented RTIs in year 2 when comparing the treated and the control group. **c** Distribution (IQR) of event RTI rate ratio (ERR, treated over control group) is plotted for the four NPI scenarios. **d** Effect Model plot for the four NPI scenarios. Each in silico clinical trial is plotted (symbols) with the number of RTIs in the control group as x-coordinate and the number of RTIs in the treated group as y coordinate. The region of clinically relevant efficacy is indicated in orange. It is defined by at least 1 prevented RTI in absolute benefit (dashed-dotted line), at least 20% reduction in number of RTIs (solid line) and at least 3 RTIs in the control group. **e** Distribution (IQR) of sample sizes per arm required to show efficacy of OM-85 treatment in reducing number of RTIs for the four NPI scenarios. **f** Distribution (IQR) of estimated patient screening times under the four NPI scenarios by assuming a hypothetical screening rate of 1000 patients per year and by taking year 2 as the selection year (without treatment). Sensitivity of these results to mechanistic uncertainty is reported in Supplementary Fig. 14.

the intervention period overlaps with the NPIs. We quantified the effect of NPIs on the efficacy of RTI prophylaxis by running in silico clinical trials using our mechanistic model applied to the oral immunomodulator OM-85 in a pediatric population suffering from recurrent RTIs. For assessing the efficacy of a prophylactic treatment absolute and relative metrics have been suggested[43,44] and therefore, we report model predictions for these metrics (Fig. 3, Methods): absolute benefit (AB, difference between rate of

RTI in both groups, Fig. 3b), event rate ratio (ERR, ratio of RTI rates between both groups, Fig. 3c), and two-dimensional analysis of rates of RTIs in treated vs. untreated patients (Effect Model, Methods: Efficacy analysis, Fig. 3d). We define here the RTI rate as the number of RTIs counted during the 12-month follow-up period (year 2 of the trial): $R_t$ for the treated group and $R_c$ for the control group. Note that $R_t$ may also refer to the (time-varying) reproduction number of an epidemic, but we refer to the rate (or

risk) of a specific event, here an RTI, in line with earlier work using the Effect Model methodology[45,46]. Additionally, we report an extended range of model predictions obtained with alternative hypotheses on key mechanisms in Supplementary Fig. 14.

To harmonize the interpretation of different efficacy metrics (see e.g., Tripepi et al.[43]), we compared the RTI rates in the treated group ($R_t$) vs. RTI rates in the control group ($R_c$) directly in a two-dimensional analysis (Effect Model, 3d). Because $R_c$ is often used to define the risk for RTI, this analysis characterizes the efficacy as a function of the risk.

The absolute benefit ($AB = R_c–R_t$) of OM-85 decreases in parallel to the reduction of the transmission rate: no reduction of the transmission rate: 1.78–2.48, 5% reduction: 1.30–1.86, 15% reduction: 0.72–1.06, and 25% reduction: 0.19–0.38 prevented RTI episodes. Assuming that an AB of 1 prevented RTI episode per year would be clinically relevant in a given context (see Discussion), only NPI-induced transmission rate reduction <15% fulfills this criterion.

The event rate ratio ($ERR = R_t/R_c$) quantifies efficacy based on event rates in the treated group relative to the control group. It is a common metric for performing statistical hypothesis testing with negative binomially distributed count data and may also be used for sample size estimations. We find that the ERR does not vary considerably in all but the strongest NPI scenario (no reduction of the transmission rate: 0.54–0.64, 5% reduction: 0.55–0.67, 15% reduction: 0.55–0.66 and 25% reduction: 0.58–0.75, Fig. 3c). In consequence, all analyses based on the ERR (i.e., sample size estimations or post-hoc power analyses) are expected to be only affected under strong NPI (e.g., strict lockdown).

In all scenarios with nonzero NPI-induced viral transmission rate reduction in year 2, virtual patients experienced fewer RTIs than in year 1 ($\geq$5; required by eligibility criterion, no lockdown in year 1). A transmission rate reduction by 5% showed a reduction of 1.1 RTIs on average (control group RTI rates are 4.0 vs. 5.1 with 5% reduction, $p < 0.001$, two-tailed Student's $t$-test). Transmission rate reduction by 15% and 25% showed a reduction of RTI rates of 2.8 and 4.3 RTIs with respect to the non-perturbed scenario (control group RTI rates are 2.3 and 0.8 vs. 5.1 with 0% reduction, both $p$-values are <0.001).

We then re-analyzed the efficacy distributions after the 12-month follow-up during the perturbed year 2 in relation to thresholds or for assumed clinical relevance (AB and $R_c$) and statistical significance of the trial (ERR) directly in the $x$–$y$ plane of Fig. 3d. We indicate a region matching three conditions (orange area in Fig. 3d, lower right quadrant): (i) recurrent RTIs (RRTIs) with >3 RTIs per year (in real-life RRTIs are often defined as more than 3 RTI episodes in the previous year and clinical benefit is considered to prevent the recurrence of RTI) as well as (ii) an absolute benefit of at least 1 RTI per year under which significant clinical benefit becomes less evident to demonstrate for such products[39] and (iii) a rate reduction of 20% in RTI rate which is a typical hypothesis for confirmatory trial design efficacy (that can be demonstrated with reasonable sample size and be clinically relevant). The percentage of in silico clinical trials complying with all three criteria is 94.0%, 0.0%, and 0.0% for the mild, medium, and strong NPI scenarios, respectively compared to 99.0% when no NPI is applied. We thus regard trials conducted as feasible when viral transmission rates are reduced by 5% but not >15%, even though they may still meet their endpoints (given that patient selection is not impaired in our simulation scenario, Fig. 3a).

**Effect of NPI scenario on recruitment**. We gauged recruitment issues for RTI prophylaxis trials with estimations of the sample size estimated for a hypothesized efficacy in a given at-risk population (as a function of NPI strength) and needed power along with a more practical time-to-recruit consideration for given eligibility criteria (Fig. 3e, f).

The sample size estimations commonly used in RTI prophylaxis trial designs are based on ERR, assuming that RTI count data are negative binomially distributed. We have therefore used a sample size estimation algorithm (Methods) using the ERR (and negative binomial dispersion coefficient) obtained from $R_c$ and $R_t$ distributions in our in silico trials for a significance level of $\alpha = 0.025$. Our sample size estimations as a function of NPI strength closely follow the trend of the ERR itself (no reduction of the transmission rate: 50, 5% reduction: 56, 15% reduction: 68, and 25% reduction: 288, Fig. 3e). Except for strong NPI, those estimates are in line with the unperturbed scenario (NPI does not affect patient selection in this example). Note that empirical power at fixed sample size of 50 patients per arm also follows this trend (no reduction of the transmission rate: 0.86, 5% reduction: 0.86, 15% reduction: 0.76, and 25% reduction: 0.34).

We estimated the time required to recruit the estimated sample sizes (Fig. 3e) if NPIs were started at the beginning of year 1 and by assuming a constant hypothetical screening rate of 1000 patients per year. Year 1 is the selection year during which patients are screened and possibly included in an in silico trial. NPIs introduced during this period could perturb the selection process. A slight reduction of the transmission rate—as small as 5%—increases the time to recruit by about 50% from 0.28 years to 0.55 years. The medium and strongest NPI scenarios (15% and 25% transmission rate reduction, respectively) lead to infeasible recruitment times (3 years and 288 years, respectively).

## Discussion

The central aspect of this work is to determine, rationalize and interpret the possible changes induced by lockdown and other non-pharmaceutical interventions (NPIs) for pandemic containment on respiratory disease trials with emphasis on RTI prophylaxis. A clinical trial has two general objectives: first, to demonstrate non-zero efficacy of the interventional strategy, a binary question with a binary answer given by a statistical test; second, to estimate the size of the clinical benefit for benefit-risk assessment. Well-designed trials fulfill both objectives through characterizing the efficacy with a quantitative measure. Not always, however, are common efficacy measures equally suitable for statistical testing and estimation of the clinical effect size. In recurrent RTIs, the event rate ratio (ERR) is often used for statistical hypothesis testing as this measure applies to negative binomially distributed count data[47,48]. Nevertheless, a measured treatment efficacy that is relative to the control group event rate is at-risk of incompletely representing the clinical benefit in case of low event rate—as in times of NPIs to mitigate the COVID-19 pandemic. We, therefore, ran in silico clinical trials (based on the SIRS model, the within-host immunological model of RTI in an individual patient, virtual population, and a simulation protocol resembling pediatric OM-85 trials) reproducing existing clinical efficacy data of OM-85 in a pediatric population suffering from RRTIs. To balance the interpretation for statistical significance versus clinical benefit considerations of these in silico trials, we applied different efficacy metrics (AB and ERR) and reconciled them in a two-dimensional analysis of treated vs. untreated rates (termed Effect Model, see Methods).

Sample size estimation is of crucial importance for planning clinical trials. For this, hypotheses on expected efficacy and chosen statistical power to detect it are needed and these may have to be adapted to the current pandemic context. Second, it is important to consider how much the efficacy in a trial can differ from an efficacy hypothesis used for the planning, especially

when perturbations arise after the trial has been planned or when sample size estimates based on historical data need to be used. Here, the post-hoc power obtained from the statistical analysis of the trial outcome might be perturbed under NPI. The analysis of the NPI-dependent efficacy of OM-85 for RTI prophylaxis revealed that the ERR remains unchanged over a broad range of NPI scenarios. Because ERR is used for statistical testing and sample size calculations, both the estimated sample size and the post-hoc power, are not substantially affected unless strong NPIs, such as strict lockdown, are applied. In such case, however, the post-hoc power of trials may be reduced for a given sample size and consequently trialists should consider an adapted efficacy scenario for obtaining more realistic estimates.

The situation is different for metrics of the clinical benefit, which assess the benefit-risk ratio. Depending on the exact context and affected population, the definition of clinical relevance may vary. For example, prophylaxis of few LRTI episodes in neonates (often associated with inception of asthma) will be clinically relevant compared to prophylaxis of a much higher number of URTIs needed for clinical relevance in pre-school children, reflecting the different effect on patients' lives and/or long-term consequences. First, children frequently suffer from RTIs (especially URTIs) and 3 RTI episodes per year can be considered a normal physiological behavior[49]. Thus, prevention of recurrence (>3) of RTIs (of which most are URTIs) appears to be clinically relevant. however, our analysis has shown that under medium and strong NPI, the annual control group RTI rate is already <3, even though it was fulfilling the definition of recurrence in the unperturbed year of patient enrollment. Second, there might be a threshold for the number of prevented events for an individual (or at the population scale), which becomes relevant from a clinical or health economic standpoint. One may assume that e.g., one prevented URTI could be regarded as relevant, but we could not identify any guidance on that topic. Here again, we found that trials under medium and strong NPI scenarios do not fulfill our criterion of AB >1 prevented RTI that could be indicative of a true clinical benefit.

The Effect Model methodology[45,46], which may be obtained from meta-analyzing existing clinical data or simulation, is a tool to rationalize control vs. treated group event rates directly. Consequently, both clinically meaningful and statistically demonstrable efficacy can be indicated in one analysis. In the optimal setting, the metrics used to demonstrate the efficacy with a statistical test goes hand in hand with the size of the effect relevant for the benefit-risk assessment. This predictivity, however, seems to be weak under pandemic conditions given the ascertained dichotomy of NPI on AB and ERR. Under the medium NPI scenario, a substantial portion of trials with positive primary endpoint evaluation could be challenged for clinical relevance of the results and, in fact, clinical benefit-related metrics seem to be the most restrictive criteria when used to assess trial feasibility a priori. We concluded from this analysis that clinical studies need to anticipate potentially weak representativity of traditional or practical endpoints for benefit-risk assessment and that either more relevant endpoints need to be chosen or feasibility studies (including computational studies such as trial simulation) should be conducted for potential trial design adjustments.

Our simulation setup for this analysis (year 1: patient selection, year 2: treatment and follow-up period) reflects RTI prophylaxis trials whose conduction takes place during the current pandemic. Therefore, we concluded that the benefit-risk assessment of these trials should account for the currently reduced disease burden, and that supporting data (such as observational studies and models) should be used to demonstrate that a low number of prevented episodes under pandemic conditions does not necessarily mean that under normal conditions equally few episodes will be prevented.

Recruitment issues are probably the earliest and a very important indicator for difficulties to conduct clinical trials in the current COVID-19 pandemic era. For respiratory disease trials, such issues may be notably due to large sample size estimates and fewer eligible patients. NPI introduced during the follow-up period, but not during the observation period, merely scales the number of prevented events in year 2 for an already recruited population (NPI not present in year 1). Therefore, the included at-risk population (nor their immunological characteristics) are not altered in such scenarios as compared to the non-perturbed one. As the ERR used for statistical efficacy testing is a metric relative to the rate of events in the control group, it is robust towards fluctuations in the overall disease burden by design. Therefore, our analysis of estimated sample size for NPI-corrected efficacy (based on event rate ratios) did not show considerably increased recruitment needs (Fig. 3e). Assuming that (e.g., for a trial with a fixed budget) an enrollment of 200 eligible patients is feasible, demonstration of efficacy in all but the strongest (25% reduction) NPI scenario, being introduced at the beginning of year 2, remains possible with the sample size planned under no-NPI scenario. We thus conclude that estimated large sample sizes and the associated issues for recruiting high numbers of patients are currently not a major difficulty for trials that have started and completed enrollment before 2020.

By considering NPI during the observational period in year 1 of a 2-year trial, we can highlight the collateral effects of COVID-19 during patient recruitment which are caused by a reduction of the size of the pool of eligible patients. At-risk populations for a given age range are included based on their history during a reference period (e.g., number of RTI episodes during the preceding 12 months), where the risk for RTI is then defined as the average number of infections per average number of viral exposures (assumed to be a constant in that time period). In practice, in trials targeting patients aged 1–6 years with recurring RTI, patients with at least 4–6 RTIs are included while the general populations suffer from e.g., only 3 episodes on average during the same time. This way of enriching the population with individuals at elevated risk, however, depends on the assumption that the virus exposure is a constant and that consequently the number of RTIs in the general population is also a constant. A reduction of the overall disease burden (e.g., by NPI), however, decreases the number of exposures and average number of RTIs in general. Consequently, in our simulations, small reductions of viral transmission already led to a reduced number of virtual patients who comply with any fixed definition of recurrent RTIs. We could translate this effect into a metric for recruitment difficulties, by considering the eligible fraction of the virtual population compared to the general virtual population and a defined fixed screening rate (Fig. 3f). Under mild NPIs, recruitment time already increased by ~50%, which questions the feasibility to recruit enough patients in time—especially for trials with a total planned duration of 6–12 months. Estimated recruitment times of 3 years for a medium NPI scenario significantly exceed the 12-month follow-up time of most trials and can thus be considered infeasible. As these analyses do not reflect any further behavioral changes and psychological effects (e.g., fear to contract COVID-19) contributing to barriers to participate in clinical trials, the presented analysis represents an optimistic scenario. Further, we did not yet account for year-to-year fluctuations in the transmission of respiratory viruses that could add to the perturbation of NPIs (or cancel it out). Nevertheless, as it is the only scenario where recruitment time does not exceed a 12-month follow-up, the mild NPI scenario is probably the only reasonable condition compatible with recruiting enough patients for RTI prophylaxis

**Table 1 Summary of the effect of NPI on clinical development by NPI strength and recommendations for each scenario. For each recommendation, a (non-exhaustive) list of specific risk mitigation measures is suggested.**

| What level of NPI is expected? | Impact on trial feasibility | Recommendation for the trialist | Specific risk mitigation measures |
|---|---|---|---|
| Weak (leading to disease burden change similar as year-to-year fluctuations) | Assessment of clinical benefit is difficult with low number of events | Reinforce and underline clinical significance of the demonstrated effect | • Select population/endpoints where a smaller (absolute) effect on RTI prophylaxis is still clinically meaningful (characterized by small minimally important difference). One example is to focus on prophylaxis of viral infection induced wheezing or asthma exacerbations, see[70,71], rather than upper RTI (mostly common cold) in the general population<br>• Comprehensive reporting of rates, relative, and absolute benefit<br>• Include secondary endpoints that add a diversified and multifaceted view to the clinical significance for assessors of the trial results (e.g., symptom-free days as RTI duration related endpoint)<br>• Seek regulator's feedback on the study protocol and statistical analysis plan with respect to clinical benefit assessment |
| Medium (leading to substantially lower disease burden; magnitude of change with respect to average exceeds year-to-year fluctuations) | Reduced post-hoc power with fixed sample size and less available patients that suffer from fixed minimum number of episodes | Mitigate loss of power through sample size adjustment, adaptive trial design, and statistical analysis tailored to rare events | • Multi-center trials with access to a larger patient pool can facilitate recruitment of larger sample sizes under difficult conditions<br>• Use Model Informed Drug Development (MIDD) to leverage the totality of evidence for an optimal trial design and extrapolation[72,73]<br>• Primary endpoint analysis based on event rate ratio (ERR) and accounting for excess zeros, e.g., zero-inflated negative binomial regression (ZINB) in frame of generalized linear models (GLM)[74,75]<br>• Use trial monitoring and (Bayesian) adaptive trial design[76] especially sample size reestimation (increasing the sample size based on interim data analysis)[77], group sequential designs[78] (trials can be stopped early once significant results are obtained, or the trial can be stopped for futility)<br>• Seek regulator's feedback on any modeling and simulation methods applied (e.g., FDA's MIDD pilot program)[79], for complex innovative trial design and the statistical analysis (e.g., FDA's complex innovative trial design pilot program[80]) |
| Strong = lockdown (leading to attenuation of seasonal epidemic) | High risk of insufficient sample size and severe recruitment issues | Change the development plan | • Change development timeline<br>• Conduct observational study to assess the effect of NPI, see e.g., ref. [81]<br>• Prioritize retrospective analyses (see ref. [82] for an example in case of OM-85).<br>• Perform exploratory modeling studies |

trials under real-world conditions. Considering that during the first UK lockdown, transmission reduction by 17.5% best reproduces the disease burden data, 5% reduction as in the mild NPI scenario is a plausible assumption for a long-term effect on viral transmission (e.g., masks, a threshold number of people in events, hand sanitizers in public places). To conclude, the selection of patients with RRTIs based on pre-pandemic historical data would only include a very small fraction of patients; thus, we suggest considering eligibility criteria tailored to the current incidence of RTIs at a given time to avoid misalignment of targeted and included population. But then, selecting the right at-risk population could become more challenging in turn.

Overall, we present here a mechanistic in silico clinical trial approach in RTI prophylaxis which can incorporate available disease burden data to output efficacy metrics relevant for assessing clinical benefits and estimating sample sizes in perturbed scenarios (or evaluating impact on the post-hoc power of a trial for a given sample size) as well as recruitment times (see summary in the first two columns of Table 1). Mechanistic description of the transmission of respiratory viruses can thereby translate lockdown and social distancing measures into a decreased rate of RTI events in patients, and into a shift of the risk-dependent efficacy for OM-85 treatment in clinical trial simulations. The selected approach has some limitations since feedback from the patient scale back to the population scale (e.g., how immunomodulation can reduce viral shedding and thus transmission) is more challenging to implement. Additionally, no data are available to calibrate OM-85's effect on viral shedding or efficacy under lockdown. We made the assumption that treatment effect and transmission are independent factors.

We highlighted that statistical significance of efficacy may be less predictive of the clinical benefit because there are fewer events to prevent (due to collateral impact of COVID-19 containment), and consequently benefit-risk assessment based on current RTI-prophylaxis trials might be difficult to establish. Recruitment of patients can be impeded as long as intermittent lockdown or

perturbations of seasonal virus transmission persist - in particular when trialists rely on pre-pandemic historical data for trial design.

Several open questions remain: What are the additional adjustments required for trial design to account for the effect of the pandemic? How does the altered and shifted seasonality of respiratory viruses affect follow-up duration? Is there a potential benefit of using inclusion criteria adapted to pandemic times (such as incidence matching), and do those adaptations risk to confound efficacy? What happens if the forecast of the disease burden turns out to be wrong?

The limitations of traditional clinical trial design methodology and the proof of concept established in this modeling and simulation study advocate for an NPI strength-dependent risk mitigation strategy for which the use of mechanistic computational models could play a pivotal role. Among measures to improve clinical significance such as balancing sample size and power, adapting statistical methods or adjusting the development plan (see Table 1 for details on suggested measures), modeling can be used to support go/no-go decisions, optimize trial design, and additionally serve as digital evidence for a wide range of RTI prophylaxis-oriented treatments.

## Methods

**Modeling approach.** The in silico clinical trials in this work are simulations performed with system models using ordinary differential equations (ODEs) embedded in a virtual population approach where parameters are described by statistical distributions rather than scalar values, in order to represent different sources of variability. Each virtual patient corresponds to a vector of parameter values drawn from the corresponding statistical distribution. Similar to a real clinical trial protocol, an in silico study protocol defines the use of the model, virtual population, simulation scenarios, and statistical analyses to answer a question of interest. It is to note that the model is rather complex thanks to its roots in systems biology and quantitative systems pharmacology. Despite the deviation from the principle of parsimony for this study, such type of models can be used for impactful (e.g., regulatory) decision making when properly validated[50].

**Multi-scale RTI disease and treatment model.** The core element of the computational approach is the coupling of a within-host mechanistic disease model, representing the viral and immune dynamics, with a between-host disease burden

model, representing the viral dynamics at the population-scale with a Susceptible, Infectious, Recovered, Susceptible (SIRS) framework, to obtain a multi-scale RTI and immunomodulation model (Fig. 1). The immunological and the SIRS models are both ODEs-based deterministic models (equations and parameters provided in the Supplementary Methods).

*Immunological within-host viral infection model.* The immunological model, implementing lytic versus nonlytic immune mechanisms during viral infection, was designed based on Wodarz et al. [51] to simulate the within-host dynamics in response to respiratory virus exposure (co-infections are not accounted) (bottom part of Fig. 1, model equations and parameters are described in Supplementary Methods, Supplementary Fig. 2, Supplementary Table 1). To translate individual occurrences of RTI events for a given patient over time into the distributions of RTI rates in the population, inter- and intra-individual variability need to be taken into account. For this, stochastic processes determine time points of viral exposure and current state of antiviral defenses. A patient-specific state of antiviral defenses (immuno-competence) is therefore distributed in the population and a layer of random fluctuations is added around each individual value. Both distributions were calibrated so that the RTI distribution in the virtual population represents a reference RTI prevalence distribution data set (obtained from a reference birth cohort[49]). Describing age as a covariate for this distribution required inclusion of a maturation term into the immune effector functions to reproduce the higher risk for RTI in young children due the still-developing immune system (Supplementary Methods, Supplementary Fig. 3).

*Between-host viral infection and disease burden model.* RTI disease burden was simulated using a SIRS model (model equations and parameters are described in Supplementary Methods: Between-host SIRS model) inspired by general literature on such models[52]. This SIRS model accounts for the seasonality of infection in an averaged manner in a given population; it is based on time-dependent transmission rates of selected viruses reproducing the seasonality of upper and lower RTIs attributed to respiratory syncytial virus (RSV), rhinovirus (RV), and influenza viruses (IV) (Supplementary Fig. 4, Supplementary Table 2). We first ran the epidemiological model alone with NPI-adjusted transmission rate (reduction of mean transmission rates $b_0$ by 0%, 5%, 15%, 25%) and compared it with data digitized from the communicable and respiratory disease reports from 2019 to 2020 published in the UK by the Royal College of General Practitioners (RCGP)[41]. The outcome of the SIRS model was then used to provide the data for the time-dependent instantaneous prevalence of RTI for the rest of the model.

*Treatment model.* To describe the immuno-modulating effect of OM-85 in RTI prophylaxis, a physiologically based pharmacokinetics and pharmacodynamics (PBPK/PD) model is linked to the immunological model through ingress in the respiratory tract of reprogrammed type-1 innate memory like cells[53], regulatory T-cells[54–56], and polyclonal IgA producing plasma cells[57,58] originating from the intestinal Peyer's patches (Supplementary Fig. 5) according to the current understanding of OM-85's mechanism of action. Implementation of administration, distribution, metabolism, and excretion follows common published approaches (Supplementary Methods: PBPK/PD model of OM-85 effect, model equations are provided as Supplementary File). In the absence of OM-85 PK data, the unknown PBPK drug-specific parameters were calibrated using rodent PK data of a similar product (OM-89)[59,60] and were allometrically scaled to human physiology (Supplementary Fig. 6, Supplementary Table 3). Unknown PD-relevant parameters were calibrated and checked using two sets of human PD response data under different treatment regimens (Supplementary Figs. 7–8, Supplementary Table 4). Calibration of remaining parameters that quantify the efficacy of OM-85 was performed based on the meta-analysis of Yin et al.[38] (Supplementary Methods: Calibration of OM-85 clinical efficacy, Supplementary Figs. 10–11).

**In silico clinical trial simulations**. We simulated placebo-controlled parallel two-arm trials of RTI prevention with OM-85 in pediatric subjects with 24-month duration (observational period of 1 year followed by a follow-up period of 1 year composed of 3 consecutive months of treatment followed by 9 without any treatment). A virtual population of 104,000 virtual patients was generated. The entire virtual population was screened during the observational period in the first year of the trial. After the first year, eligibility criteria were evaluated and randomization was performed. In line with the range of annual RTI episodes typically defining RRTIs (3–6), children that experienced at least 5 RTIs were included into the follow-up period. Included virtual subjects were randomly allocated with equal weight to the interventional and control arms. During the first 3 months of the follow-up period, OM-85 was administered every day during the first 10 consecutive days of each month, in line with the currently approved dosing regimen of OM-85 in the prevention of RTIs. The primary outcome was the number of RTIs during 1 year follow-up, which was assessed at the end of the trial. 600 in silico clinical trials were simulated for each of the 4 different NPI scenarios (reduction of the transmission rate by 0, 5%, 15%, 25%) by randomly sampling 50 subjects per arm from the screened virtual population (studies meta analyzed by Yin et al.[38] have enrolled in average 45.4 patients per arm).

**Efficacy analysis**. The Effect Model approach[45,61] is a tool, which relates the rates (or risks) of events without treatment (Rc) and with (Rt), as supported by empirical evidence, simulations, and theoretical considerations[62–65]. While simulations can be conducted for the same patient in different arms in in silico trials and yield paired observations, the Effect Model can also be reconciled with meta-analyses[62,66]. Here we have used a similar approach that compares RTI rates in a series of individual in silico clinical trials, thus not reporting individual, but risk-stratified group metrics. As efficacy metrics, we consider absolute benefit (AB) and the event rate ratio (ERR). Average ERR and AB were assessed at 1 year follow-up. AB determined from a single in silico trial is the arithmetic difference between mean RTI rate in the control group (Rc) and in the treatment group (Rt). ERR refers to the ratio Rt/Rc. Distribution of the ERR and AB per scenario contain pooled results for different mechanistic conditions and is visualized as the maximum interquartile range (defined as the difference between maximum 75th percentile and minimum 25th percentile across mechanistic conditions). AB and ERR were analyzed with a paired *t*-test/ANOVA with α level set at 0.05.

**Sample size and recruitment estimation**. Sample size calculations for primary endpoint analyses of RTI prophylaxis trials require an adapted statistical method for overdispersed count data. We performed generalized linear regression analysis with negative binomial distributions (mean and dispersion parameter, *glm* function of the R package MASS 7.3-55) for the subsequent use of these parameters in the sample size calculation method proposed by Zhu et al. [67]. Calculations employed the power.nb.test function of the MKmisc package (1.8) given the ratio of rates in both trial arms, average dispersion parameter between both arms (thus varied per NPI scenario; note that fixed dispersion parameter gives similar results, Supplementary Fig. 15), an α of 0.025 and correction for average study duration (e.g., due to dropout $\kappa = 0.75$). Based on sample size calculations and the fraction of the entire virtual population eligible for inclusion, time to recruitment was calculated, assuming that in a typical study in respiratory diseases, a screening rate of 1000 patients per year can be achieved per center.

**Reporting summary**. Further information on research design is available in the Nature Research Reporting Summary linked to this article.

## Data availability

The data (simulation outputs, in silico clinical trials and analyses) generated in this study and needed to reproduce the results presented in the figures are provided as comma-separated-value files (csv) compressed into one file (zip) as Supplementary File. 5-year average and the 2019–2020 upper and lower RTI (URTI and LRTI) incidence from the communicable and respiratory disease report 2019–2020 published in the UK by the Royal College of General Practitioners (RCGP) can be accessed at the following address: https://www.rcgp.org.uk/-/media/Files/CIRC/WeeklyReport_Summer_wk31_2020.ashx. All other datasets used (viral load evolution after experimental viral challenge[68], PK/PD data[58–60,69], age-dependent RTI distribution[49], meta-analyzed clinical efficacy[38]) were obtained from published reports whose references are cited. Source data are provided with this paper.

## Code availability

The model source is provided in the Supplementary File as an SBML file (Level 3 Version 2) along with a Python (3.7) script to run the model for a reference patient in a reference scenario using libroadrunner (http://libroadrunner.org/) a C/C++ library that supports simulation of SBML based models (version 2.2.0), reference model outputs for all variables (.csv, .pdf), summary model implementation table (.xls) as well as mapping between the source and the human-readable description (pdf, xls). For the analysis, the code (Python 3.9.7, Numpy 1.19.5, Scipy 1.8.0, rpy2 3.4.5, pandas 1.4.1, Matplotlib 3.5.1, JupyterLab 3.2.9, R 4.0.4, MKmisc 1.8, MASS 7.3-55) used to produce the figures is provided along with the necessary data as Supplementary File.

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

## Acknowledgements

We acknowledge feedback by Jim Bosley and Shiny Martis B. on the manuscript draft and technical support from Eliott Tixier, Roman Cheplyaka, and Louis Philippe.

## Author contributions

A.K., S.A., L.L., C.P., and A.C. supervised the study. S.A., C.C., I.F., N.G., S.G.N., D.S. developed the model, performed simulations, and analyzed the results. S.A. and A.K. wrote the manuscript. All authors contributed to the discussion of the results and reviewed the manuscript.

## Competing interests

A.K., S.A., C.C., I.F., N.G., S.G.N., D.S., R.K., B.I., J.P.B. are employees of Novadiscovery. A.C., L.L., C.P. are employees of OM Pharma. Novadiscovery and OM Pharma funded the study.
