## [Peer Review File · Nature Communications]

Reviewers' Comments:

Reviewer #1:

Remarks to the Author:

In this manuscript, Arsène and colleagues use a combined within- and between-host respiratory tract infection (RTI) model to illustrate important considerations for designing and interpreting clinical trials for RTI therapeutics in the context of perturbed RTI circulation due to non-pharmaceutical interventions (NPIs) during the COVID-19 pandemic. They find that, even if trial's main endpoints are attained, NPIs can yield an apparently lower clinical efficacy and cost effectiveness, potentially leading to the rejection of good drug candidates. The authors also demonstrate that NPIs can increase the necessary number of trial participants and the amount of time needed to recruit them, rendering some trials infeasible that would be possible during periods of normal RTI circulation.

As far as I can tell, the authors' approach is appropriate, theoretically sound, and well-validated. My experience is mainly with population-level disease transmission models, and so I was unable to rigorously assess the validity of the within-host immunological and PK/PD models, though they appear reasonable and match well with the available empirical data. The population-level model is far simpler than the within-host model, but nevertheless I believe this is appropriate for a study that aims to illustrate broad principles, and again the between-host model is well validated against empirical data.

Overall, I believe this is a useful and important study, illustrating a set of clinically important points that might otherwise be overlooked. My comments are minor.

ABSTRACT

Here, and in various other places (e.g. Lines 52 and 147), the authors refer to "knowledge-based mathematical modeling". I'm not sure what this means. Is it synonymous with "mechanistic"? If so, I'd suggest using the more precise term.

RESULTS

95-96: I'd appreciate a fuller description here of how the ERR varies between NPI scenarios. Based on these numbers and Figure 3, it appears that stricter lockdowns yield a higher mean and upper 75% quantile for the ERR, but the lower 25% quantile is pretty consistent across scenarios. What can we learn from this?

DISCUSSION:

142: I think that "probability density" isn't quite the right term here; presumably the integral of the SIRS model's outcome can sum to a number greater than 1, especially when re-infections are possible. I think that this is actually an instantaneous hazard of infection, in the survival-analytic sense of the word.

134-153: This entire section (from the beginning of the Discussion through "(e.g. to another NPI scenario)") seemed superfluous; pretty much all of this information has been presented already, and anything that's new would fit better in a Methods section. I do think that the limitations (beginning with "As a limitation of the chosen approach...") are good and should be included somewhere later in the Discussion. Then, I'd suggest starting the Discussion with what's currently the second paragraph ("The central aspect of this work..."). This would help to shorten what is already a long Discussion and would start the Discussion with the most relevant take-aways from the analysis.

METHODS:

316: "more than 100,000" - why not give the actual number (and why not exactly 100,000)?

I like the figures - Figure 1 was especially helpful.

SUPPLEMENT:

127: I don't think λ has been defined, and it should be here. Presumably it's the integral of $p(t)$ between t_0 and t_A ?

135: The number 10 for gamma should have units (presumably days -- and if so, should this be 1/gamma?)

Figures S10-11 - can you add a legend describing the colors here? I know you reference them in the text of the Supplement but it would be a lot nicer to be able to match them up right here.

Reviewer #2:

Remarks to the Author:

General Summary:

This is a well-designed simulation study that examines the effect of COVID-19 NPIs on various results from clinical trials of pediatric patients with RRTIs. The underlying models appear sound and are described clearly. The results point to two key findings: (1) that trial results, especially absolute benefit effect measures, can be substantially affected by such NPIs; but (2) relative effect measures are unlikely to be substantially affected unless the NPI effect is strong. This has implications for sample size and the feasibility of the trial, as the authors explore.

They key missing piece to me is some indication of how trial investigators should use these findings. Is there a metric that will have some generalizability (i.e., beyond just the specific scenario of RRTI transmission modelled here) that trialists can use to adjust sample sizes a priori or mid-trial if NPIs are enacted? Or that can be used to relate absolute effect sizes identified under different NPI conditions? While the findings presented here are relevant on their own, such a metric would significantly increase the impact of this work.

Major Comments:

- P. 2, ll. 70–72: As I understand it, the model is calibrated to match observational data of RTI incidence. Given the effects of the pandemic on health care visits, this might reflect reduced true incidence and reduced testing for RTIs. In a randomized trial with specific follow-up plans, however, that reduced testing may not occur. So the underlying incidence data used here may differ from what would be seen in randomized trials. I think some comments about this (either here or in the discussion) would be warranted, as it could affect interpretation of the results, or the authors can argue it would not have a substantial effect.
- P. 3, ll. 88–92: The IQRs used should be explained here. I find the maximum IQR concept rather difficult to interpret, especially as the 75th and 25th percentiles shown may be from different mechanistic models. I think the results would be more interpretable if one main model were chosen, and the others shown as sensitivity analyses in the supplement, so that the effects of different modeling assumptions could be seen.
- P. 3, l. 93: Relatedly, it would be very useful if the results presented included some benchmark for the “true” effect of OM-85 in the model. Given the mechanistic model, there is not one effect, but the average or expected reduction in transmission may be able to be found from the model (or at least some summary of the studies that are used to calibrate the model parameters). That would improve interpretation greatly.
- Fig. 3: In the discussion, the authors cite hypothesis testing as the main value of the RTI rate ratio effect size. Given that and the importance of hypothesis testing in such trials, the empirical power of the trials should be displayed in the main text.
- Fig. 3c: I would like to see some discussion of the increased (maximum 75th percentile) RTI rate ratio observed under the strong NPI. Do the authors think this is likely an outlier simulation result, or is there some reason that might occur?
- P. 3, l. 124: is the dispersion coefficient that is used estimated from the no-reduction scenario in all cases or is it varied by transmission reduction scenario? The former is more realistic for trial planning, since investigators are more likely to have detailed information on counts in non-NPI settings than NPI settings. My guess is using the former will lead to more extreme sample size differences than the latter, but it would be interesting to know how the dispersion coefficient changes with increased NPI strength.
- Discussion: the numerical results presented are very specific to the setting under examination. As stated above, identifying some generalizable method for adjusting sample size/effect size would be very valuable. If that is not possible, however, the authors should clearly state this limitation

and the key factors that may change the magnitude of the discrepancy in effect sizes under NPIs.

Minor Comments:

- P. 2, ll. 37–40: This sentence seems to be cut off in the middle; I also don't understand what is meant by "respiratory disease prophylaxis and international trials". I expect that all trials, whether within one country or multi-site trials across countries, will be affected.
- Fig. 2: Some mention should be made of the relatively poor fit of the model in January–March, and whether this raises concern for the usefulness of the results.
- P. 3, ll. 79–80: It would be useful here to clarify that the intervention/follow-up period of the trial is only the second year (coincident with the NPIs).
- Fig. 3f: This is not very useful with the large y-axis. I think this panel would be improved by showing only the 0, 5, and 15% reduction scenarios.
- P. 3, ll. 99–106: I think it would be easier to interpret the other results if the reduction between Year 1 and Year 2 in R_c were introduced first, so I recommend moving this paragraph above the previous two.
- P. 3, ll. 109–114: Can the authors cite guidelines or example trials that use these benchmarks for clinical relevance?
- P. 3, l. 128: Why is the NPI now taken to begin in year 1? Clarified in discussion, but should be mentioned here.
- P. 4, ll. 155–157: Can you clarify these sentences a bit? Is this saying that the model assumes that OM-85 only affects probability of infection of the individual and not their probability of onward transmission?

Reviewer #3:

Remarks to the Author:

This is a novel study introducing one in silico approach to replace clinical trials of respiratory tract infections taking into consideration what the impact of a pandemic can be on their feasibility. I have some concerns for this submission.

- The authors need to decide what they want the outcome of their study to be. It is obvious that they are concerned about the feasibility of running a trial for respiratory tract infections (RTI) taking into consideration the restrictions imposed by COVID-19 pandemic. However they introduce the OM-85 immunomodulator making the reader wondering what is the message the authors wish to confer: provide the efficacy of OM-85 to prevent RTIs; cure RTIs or to predict how long it will take to finalize a study.
- The study is inconclusive. The authors feed their program with assumptions which are based on the COVID-19 pandemic trying to interpret it with known epidemiology of non-COVID infections.

Reviewer #4:

Remarks to the Author:

Arsene et al propose a model to predict the impact of Covid-19 on the design of clinical trials of drugs against other respiratory viral diseases. Their starting assumption is that lockdown and other NPI during Covid-19 has led to a change in the epidemiological patterns of respiratory viral diseases, leading to less contacts and less infections, and that this needs to be accounted for when evaluating drugs against viral infections. To address this question the authors use a within-host and between-host model that connects the time-varying prevalence of the infection in the population with the individual risk of developing the infection, and they study the impact of different reductions of virus prevalence on different clinical outcome.

Although the model is conceptually interesting, the implications are fairly limited. There is not much surprise to find that drastic reductions in contacts during lockdowns will reduce the risk of infection, and hence will affect the absolute benefit of RTI interventions as well as the enrollment time. Likewise, it is expected that relative benefits will not be much affected. Moreover, the authors rely on a very specific intervention, OM-85, which further narrows the application of their study. Overall I find the model conceptually interesting, but with very limited implication for

biology or clinical development.

I have some more specific comments:

1) the within-host model is very complicated, and it is not clear at all how these parameters were "calibrated". At least some sensitivity analyses would be needed

2) Since the model not virus specific, I would add that it is not clear why parameters would be similar for different viruses. When looking at viruses for which viral kinetic modeling has been done, parameters are fairly different between SARS-CoV-2, Influenza or RSV

3) It is unclear how infection acquisition is considered. The authors mention a "detectable RTI" but it is unclear how this is related to the within-host model.

4) If I were in charge in designing such a clinical trial, I think that I would use as a primary or secondary endpoint a factor related to symptom (severity, duration). I would suspect much more statistical power to detect a drug effect on these factors than on the acquisition. This could be an interesting aspect to consider in such a detailed analysis on designing clinical trials in RTI.

Novadiscovery SA

Alexander Kulesza, PhD

Team Leader | Biomodeling and Simulation

1 Place Verrazzano, 69009 Lyon, France

+33 7 82 92 44 620

alexander.kulesza@novadiscovery.com

novadiscovery.com

Lyon, February 7th 2022

Manuscript Revision

Dear Reviewers,

Thank you for your feedback on our manuscript and the comments and suggestions. We would like to address these point-by-point as detailed below.

All respective changes in the manuscript are highlighted in red for addition / modification and strike-through for deletion. We additionally provide a clean version.

Reviewer #1	2
Reviewer #2:	5
Reviewer #3:	18
Reviewer #4:	20

Reviewer #1

ABSTRACT

Here, and in various other places (e.g. Lines 52 and 147) , the authors refer to "knowledge-based mathematical modeling". I'm not sure what this means. Is it synonymous with "mechanistic"? If so, I'd suggest using the more precise term.

Response 1.1

We initially intended to make the distinction between purely mechanistic models that in principle are only physics based (diffusion, advection) or rely on chemical kinetics. Typically, mechanistic models must be used with assumptions and simplifications backed up by the knowledge that these do not alter the representativity of the system. We agree that this new terminology might be confusing and stick to "mechanistic" throughout the paper.

The third sentence of the abstract now reads: "Clinical trial design based on pre-pandemic historical data therefore needs to be put in question. In this article, we show how mechanistic mathematical modeling can be used to address this issue."

We removed "knowledge-based" from Line 52.

The sentence Line 147 now reads: "We calibrated the SIRS model with heterogeneous viral prevalence data in a mechanistic manner, which limits the degrees of freedom to calibrate the model to the data."

RESULTS

95-96: I'd appreciate a fuller description here of how the ERR varies between NPI scenarios. Based on these numbers and Figure 3, it appears that stricter lockdowns yield a higher mean and upper 75% quantile for the ERR, but the lower 25% quantile is pretty consistent across scenarios. What can we learn from this?

Response 1.2

We thank the reviewer for pointing out this relevant observation. The fact that the lower quartile for strong NPI scenario remains at the same level as for the other NPI scenarios is primarily the result of the chosen sample size being too small compared to the reduced effect size (increased ERR) with strong NPI. If the sample size is increased to the unrealistic value of 500 patients per arm, then the increased ERR (so decreased effect size) is better captured, and the lower 25% quantile is also increased with strong NPI (see Figure below).

The ERR is universally increased at all strong NPI scenarios due to the count nature of the data. At very low RTI incidence a significant part of the sample consists of observations with 0 RTIs and thus the treatment effect cannot be observed.

DISCUSSION:

142: I think that "probability density" isn't quite the right term here; presumably the integral of the SIRS model's outcome can sum to a number greater than 1, especially when re-infections are possible. I think that this is actually an instantaneous hazard of infection, in the survival-analytic sense of the word.

Response 1.3

We agree with the reviewer on the fact that a probability density would imply the integral to be 1. At the same time, we are discussing here more the likelihood of being exposed to an infected individual rather than of being directly infected. Thus, in line with your recommendation, we now utilize "instantaneous hazard of infection".

Line 77-79 now reads: "Supported by this agreement, we applied this epidemiological model to modulate the instantaneous hazard of exposure to infection to RTI-causing viruses in our in silico trials with four different NPI scenarios."

134-153: This entire section (from the beginning of the Discussion through "(e.g. to another NPI scenario)") seemed superfluous; pretty much all of this information has been presented already, and anything that's new would fit better in a Methods section. I do think that the limitations (beginning with "As a limitation of the chosen approach...") are good and should be included somewhere later in the Discussion. Then, I'd suggest starting the Discussion with what's currently the second paragraph ("The central aspect of this work..."). This would help to shorten what is already a long Discussion and would start the Discussion with the most relevant take-aways from the analysis.

We thank the reviewer for this suggestion which we followed. The Discussion now starts with "The central aspect of this work.." and the limitations have been integrated into a later part of the Discussion.

The paragraph starting with "Overall, we present here a mechanistic.." now ends with: "The selected approach has some limitations since feedback from the patient scale back to the population scale (e.g. how immunomodulation can reduce viral shedding and thus transmission) is more challenging to implement. Additionally, no data are available to calibrate OM-85's effect on viral shedding or efficacy under lockdown. We made the assumption that treatment effect and transmission are independent factors."

Response 1.4

METHODS:

316: "more than 100,000" - why not give the actual number (and why not exactly 100,000)?

Response 1.5

We simulated 2,000 virtual patients per NPI scenario for the 12 mechanistic scenarios + the untreated scenarios so exactly 104,000 in total. We now indicate the exact number of simulations in the manuscript.

I like the figures - Figure 1 was especially helpful.

SUPPLEMENT:

127: I don't think lambda has been defined, and it should be here. Presumably it's the integral of p(t) between t_0 and t_A?

Response 1.6

Here, $\lambda = \gamma * (\text{integral of } p(t) \text{ between } t_0 \text{ and } t_A) / (\text{integral of } p(t) \text{ over 1 year})$. Note that this is a result of the MC procedure and not an input of it. We have added this to the Supplementary (Line 129, Section "Interface of between host and within host model of viral infection")

135: The number 10 for gamma should have units (presumably days -- and if so, should this be 1/gamma?)

Response 1.7

It corresponds to a number of events (viral exposure) and thus does not have units.

Figures S10-11 - can you add a legend describing the colors here? I know you reference them in the text of the Supplement but it would be a lot nicer to be able to match them up right here.

Response 1.8

We followed the reviewer's suggestion and added a Supplementary Figure (S12) with the color coding for the different model hypotheses and referenced it in the captions of Figures S10-11.

Figure S12: Color code for the different combinations of the two hypotheses on dendritic cells sensitivity (x-axis; very low: 0.01, low: 0.05, medium: 0.1 and high: 0.2) and on lifetime of type-1 innate immune cells (y-axis; short: 30 days, medium: 60 days, long: 90 days).

Reviewer #2:

General Summary:

They key missing piece to me is some indication of how trial investigators should use these findings. Is there a metric that will have some generalizability (i.e., beyond just the specific scenario of RRTI transmission modelled here) that trialists can use to adjust sample sizes a priori or mid-trial if NPIs are enacted? Or that can be used to relate absolute effect sizes identified under different NPI conditions? While the findings presented here are relevant on their own, such a metric would significantly increase the impact of this work.

Response 2.1

We would like to express our gratitude to Reviewer 2 for pointing out that the initial manuscript left gaps in the conclusions. We took the opportunity to analyze the direct outcome of the simulation study, especially the main effect of weak, medium and strong NPI on the trials and recommendations for the trialist to ameliorate these alterations. We introduce Table 1 highlighting, summarizing and detailing issues and recommendations alongside specific risk mitigation measures. The table reads as follows.

Table 1. Summary of the effect of NPI on clinical development by NPI strength and recommendations for each scenario. For each recommendation, a (non-exhaustive) list of specific risk mitigation measures is suggested.

What level of NPI is expected?	Impact on trial feasibility	Recommendation for the trialist	Specific risk mitigation measures
Weak (leading to disease burden change similar as year-to-year fluctuations)	Assessment of clinical benefit is difficult with low number of events	Reinforce and underline clinical significance of the	 Select population/endpoints where a smaller (absolute) effect on RTI prophylaxis is still clinically meaningful (characterized by small "minimally important difference"). One example is to focus on prophylaxis of viral

		demonstrated effect	infection induced wheezing or asthma exacerbations, see \cite{10.1016/j.iac.2008.03.001,10.1016/S0140-6736(10)61380-3}, rather than upper RTI (mostly common cold) in the general population.  • Comprehensive reporting of rates, relative and absolute benefit • Include secondary endpoints that add a diversified and multifaceted view to the clinical significance for assessors of the trial results (e.g. symptom-free days as RTI duration related endpoint) • Seek regulator’s feedback on the study protocol and statistical analysis plan with respect to clinical benefit assessment
Medium (leading to substantially lower disease burden; magnitude of change with respect to average exceeds year-to-year fluctuations)	Reduced post-hoc power with fixed sample size and less available patients that suffer from fixed minimum number of episodes	Mitigate loss of power through sample size adjustment, adaptive trial design and statistical analysis tailored to rare events	 ▪ Multi-center trials with access to a larger patient pool can facilitate recruitment of larger sample sizes under difficult conditions ▪ Use Model Informed Drug Development (MIDD) to leverage the “totality of evidence” for an optimal trial design and extrapolation \cite{10.1002/cpt.2422,10.1183/13993003.congress-2021.PA3152} ▪ Primary endpoint analysis based on event rate ratio (ERR) and accounting for excess zeros, e.g. zero inflated negative binomial regression (ZINB) in frame of generalized linear models (GLM) \cite{10.3844/amjbsp.2017.1.12,10.1007/978-0-387-21706-2} ▪ Use trial monitoring and (Bayesian) adaptive trial design \cite{10.1056/nejmra1510061} especially sample size reestimation (increasing the sample size based on interim data analysis) \cite{10.3414/ME09-02-0060}, group sequential designs \cite{10.1177/0962280218773115} (trials can be stopped early once significant results are obtained, or the trial can be stopped for futility) ▪ Seek regulator’s feedback on any modeling and simulation methods applied (e.g. FDA’s MIDD pilot)

			program)\cite{MIDD}, for complex innovative trial design and the statistical analysis (e.g. FDA's complex innovative trial design pilot program\cite{CITD}))
Strong = lockdown (leading to attenuation of seasonal epidemic)	High risk of insufficient sample size and severe recruitment issues	Change the development plan	 ▪ Change development timeline ▪ Conduct observational study to assess the effect of NPI, see e.g. \cite{10.1016/s2468-2667(20)30090-6} ▪ Prioritize retrospective analyses (see \cite{10.3390/ijerph18136871} for an example in case of OM-85) ▪ Perform exploratory modeling studies

To support the recommendations and point the reader to a more detailed justification we have added the following references to the table

- Carroll et al. (2008) The Impact of Respiratory Viral Infection on Wheezing Illnesses and Asthma Exacerbations
DOI: 10.1016/j.iac.2008.03.001
- Busse et al. (2010) Role of viral respiratory infections in asthma and asthma exacerbations
DOI: 10.1016/s0140-6736(10)61380-3
- Drazen et al. (2016): Adaptive Designs for Clinical Trials,
DOI: 10.1056/nejmra1510061
- FDA (2018) Model Informed
- FDA (2018) Complex Innovative Trial Design Pilot Meeting Program
fda.gov/drugs/development-resources/complex-innovative-trial-design-pilot-meeting-program
- Schmidli et al. (2010) Blinded Sample Size Reestimation with Negative Binomial Counts in Superiority and Non-inferiority Trials
DOI: 10.3414/me09-02-0060
- Mütze et al. (2018): Group sequential designs for negative binomial outcomes, DOI: 10.1177/0962280218773115
- Lukusa (2017) Review of Zero-Inflated Models with Missing Data
DOI: 10.3844/amjbsp.2017.1.12
- Venables (2002) Modern Applied Statistics with S
DOI: 10.1007/978-0-387-21706-2
- Cowling (2020) Impact assessment of non-pharmaceutical interventions against coronavirus disease 2019 and influenza in Hong Kong: an observational study
DOI: 10.1016/s2468-2667(20)30090-6
- Arsène et al. (2021) Abstract: Mechanistic model based meta-analysis for paediatric respiratory tract infection prophylaxis trial design DOI:10.1183/13993003.congress-2021.pa3152
- Cantarutti et al. (2021) Use of the Bacterial Lysate {OM}-85 in the Paediatric Population in Italy: A Retrospective Cohort Study
DOI:10.3390/ijerph18136871

For the integration of this table, we made the following changes to the final sections of the Discussion:

Overall, we present here a mechanistic *in silico* clinical trial approach in RTI prophylaxis which can incorporate available disease burden data to output efficacy metrics relevant for assessing clinical benefits and estimating sample sizes in perturbed scenarios (or evaluating impact on the post-hoc power of a trial for a given sample size) as well as recruitment times (see summary in the first two columns of Table 1).

Mechanistic description of the transmission of respiratory viruses can thereby [...].

Several open questions remain: What are [...].

The limitations of traditional clinical trial design methodology and the proof of concept established in this Modeling & Simulation study advocates for an NPI strength-dependent risk mitigation strategy for which the use of mechanistic computational models could play a pivotal role.

Among measures to improve clinical significance such as balancing sample size and power, adapting statistical methods or adjusting the development plan (see Table 1 for details on suggested measures), modeling can be used to address these questions in detail, to support go/no-go decisions in clinical development, optimize trial design, and additionally serve as digital evidence for a wide range of RTI prophylaxis-oriented treatments.

Furthermore, when trials are deemed infeasible, the models could harness RTI disease burden monitoring (and prediction of COVID-19 and its associated containment measures) to indicate the time point when delayed or stopped trials can be restarted.

Major Comments:

- P. 2, ll. 70–72: As I understand it, the model is calibrated to match observational data of RTI incidence. Given the effects of the pandemic on health care visits, this might reflect reduced true incidence and reduced testing for RTIs. In a randomized trial with specific follow-up plans, however, that reduced testing may not occur. So the underlying incidence data used here may differ from what would be seen in randomized trials. I think some comments about this (either here or in the discussion) would be warranted, as it could affect interpretation of the results, or the authors can argue it would not have a substantial effect.

Response 2.2

We acknowledge the reviewer's question whether a reduced RTI incidence during pandemic times is primarily due to social distancing measures or a result of detection bias due to significantly reduced testing.

It is likely that during Covid-19 infection peaks hospitals initially reduced testing for other viruses than SARS-CoV-2. The RCGP (Royal College of General Practitioners) data that we reported in this manuscript are based on weekly reports from the RCGP Research and Surveillance Centre. The latter is "an active research and surveillance unit that collects and monitors data from over 1700 practices across England and Wales". Thus, it represents a network of diagnoses made by various general practitioners. Hence, we think that this monitoring remained accurate during the pandemic for measuring RTI incidence though we recognize that there may have been reduced

access to general healthcare during strict lockdowns. However, the sharp decline in RTI incidence is confirmed by other sources (which we detailed below), which is why we think that reduced testing cannot be a major cause for the decreased RTI incidence during the pandemic.

Other sources worldwide with access to reports of GPs and hospitalization outline the same trends over time and place – a substantial, sharp decline in RTI incidence since the start of the pandemic (Wang et al., 2021, 10.3389/fped.2021.584874; Tanislav and Kostev, 2022, 10.1002/jmv.27321; Olsen et al., 2021, 10.15585/mmwr.mm7029a1). RSV and influenza viruses are spread primarily by droplet transmission similar to SARS-CoV-2. Therefore, the effect of social distancing measures on RTI is biological plausible, although causality cannot directly be inferred. Wang et al. conducted a retrospective observational study in China, where the study population was selected from hospitalized children with respiratory diseases. As China was officially not concerned by the Covid-19 pandemic after the initial outbreak in Hubei and the subsequent strict NPIs, the hospitals in China were allegedly not overwhelmed. Therefore, the strong decline in hospitalized patients due to respiratory pathogens (see figure below, also shown below this paragraph) from February 2020 is most likely attributable to general RTI incidence decrease due to NPIs and not due to reduced testing and full hospitals.

FIGURE 1 | Infection of four major respiratory viruses in children of different ages from January 1, 2018 to January 31, 2021.

Tanislav and Kostev analyzed German electronic medical record data, which were obtained by processing and anonymizing drug prescriptions, diagnoses, and general medical and demographic data obtained in the practices of GPs and specialists. It is representative of 3% of all private practices in Germany and proved to be a population-representative database. As these data are similar in the acquisition to the RCGP data from UK and reports and show a similar decline of a great majority of RTIs (see Table 2, also shown below this paragraph), we consider this as confirmatory of the RCGP data.

TABLE 2 Total annual change in infection diagnoses (per practice) in general and pediatrician practices (April 2019–March 2020 compared to April 2020–March 2021)

Diagnosis	General practitioners			Pediatricians		
	April 2019–March 2020 mean (SD)	April 2020–March 2021 mean (SD)	Change in %	April 2019–March 2020 mean (SD)	April 2020–March 2021 mean (SD)	Change in %
Viral or unspecified gastrointestinal infections	192.4 (161.0)	116.2 (102.5)	-40***	321.8 (192.4)	138.3 (96.9)	-57***
Respiratory tract infections						
Acute nasopharyngitis	52.2 (122.6)	33.2 (72.1)	-36	188.5 (326.3)	133.9 (237.1)	-29
Acute sinusitis	40.3 (66.6)	16.9 (27.5)	-58***	12.8 (25.9)	4.5 (9.8)	-66***
Acute pharyngitis	62.5 (103.2)	31.5 (86.9)	-50***	173.1 (228.8)	87.2 (137.1)	-50***
Acute tonsillitis	57.5 (60.5)	24.3 (26.4)	-58***	250.9 (226)	87.9 (120.3)	-65***
Acute laryngitis	27.0 (50.7)	9.7 (24.3)	-64***	96.4 (169.8)	36.0 (100.2)	-63***
Acute upper respiratory infections of multiple and unspecified sites	326.7 (283.3)	260.8 (346.3)	-20***	798.6 (606.9)	517.5 (484)	-35***
Influenza	23.7 (72.7)	6.7 (27.2)	-71***	54.2 (76.7)	5.2 (22.6)	-90***
Viral pneumonia	0.3 (2.8)	1.0 (4.4)	+229***	1.4 (4.0)	0.4 (1.2)	-73***
Other pneumonia	28 (28.5)	14.3 (13.6)	-49***	71.2 (77.9)	16.9 (25.2)	-76***
Acute lower respiratory infections	143 (176.2)	54.5 (132.4)	-62***	233.4 (257.9)	88.8 (149.5)	-62***

***<0.001.

In Olsen et al., the Centers for Disease Control and Prevention (CDC) analyzed virological data from US laboratories available through the US World Health Organization Collaborating Laboratories System and CDC’s National Respiratory and Enteric Virus Surveillance System (NREVSS). Reporting bias via the participating laboratories was excluded due to several prerequisites, hospitalization rates were calculated via accessing several Hospitalization Surveillance Networks (HSN) for different respiratory pathogens. Figure 1 (see below this paragraph) in their publication shows an abrupt reduction in the rate of positive tests and hence in the incidence of the most prevalent respiratory pathogens – influenza and RSV (Pattimore and Jennings 2009, 10.1016/b978-032304048-8.50035-9) – and a notable reduction in adenovirus and rhinovirus incidence despite the number of tests executed in the time window of 2021 until mid-2021 comparable to the one before the pandemic.

In total, these studies are consistent in reporting a sharp decline in RTI with various methods of monitoring and in various conditions. We thus believe that the drop in RTI incidence reported by the RGCP data can be majorly attributed to NPIs and not reduced testing.

We have added those references in the main text of the manuscript, Line 32 now reads: "Detection bias due to reduced testing is not a significant confounder as several sources worldwide [0.1002/jmv.27321, 10.3389/fped.2021.584874, 10.15585/mmwr.mm7029a1] reported a sharp decline in the number of RTIs relative to the number of tests."

- P. 3, ll. 88–92: The IQRs used should be explained here. I find the maximum IQR concept rather difficult to interpret, especially as the 75th and 25th percentiles shown may be from different mechanistic models. I think the results would be more interpretable if one main model were chosen, and the others shown as sensitivity analyses in the supplement, so that the effects of different modeling assumptions could be seen.

Response 2.3

We are thankful to the reviewer for this suggestion and accordingly changed Figure 3 to display the results obtained with the reference modelling assumptions (main mechanistic model) and

added Supplementary Fig S14 to report sensitivity of these results to the set of modeling assumptions.

Supplementary Figure S14: Sensitivity of main results to mechanistic uncertainty. For this, we simulated different mechanistic scenarios in parallel. We used 12 different conditions (testing different immunogenic hypotheses on the effect of OM-85, Supplementary Methods: Mechanistic uncertainty management) for each of the 4 non-pharmaceutical interventions (NPIs) scenarios and pooled the results. Each mechanistic scenario is color coded (Figure S12). Similarly to Figure 3 of main text, we ran *in silico* clinical trials with 4 scenarios of NPIs with increasing strength (absent, mild, medium and strong) modelled by a decrease of the transmission rate parameter (no reduction, -5%, -15% and -25%, respectively). There is no NPI during year 1. The NPIs are started at the beginning of year 2 as well as the treatment (10 daily administrations of 3.5 mg of OM-85 from the beginning of the month for 3 consecutive months). Number of RTIs are counted for the complete duration of year 2. For the 12 mechanistic scenarios and for the 4 NPI scenarios, we report: a) the distribution (IQR) of absolute benefit; b) the distribution (IQR) of event RTI rate ratio (ERR, treated over control group); c) the distribution (IQR) of sample sizes per arm required to show efficacy of OM-85 treatment in reducing the number of RTIs; d) the distribution (IQR) of estimated patient screening times assuming an hypothetical screening rate of 1,000 patients per year and by taking year 2 as the selection year (without treatment).

In Supplementary Figure S14, we can observe that absolute benefit is comparable for all NPI scenarios throughout the range of mechanistic hypotheses whereas ERR and consequently sample size and recruitment times are more affected (especially with the strong NPI scenario). Interestingly, lifetime of type-1 activated innate immune cells is the most determinant factor where short-lived cells reduce the effect of the strong NPI scenario more compared to long-lived cells.

We now reference this Figure in the Results section of the main text, Line 89-90 reads: " Additionally, we report an extended range of model predictions obtained with alternative hypotheses on key mechanisms in Supplementary Figure S14."

- P. 3, l. 93: Relatedly, it would be very useful if the results presented included some benchmark for the “true” effect of OM-85 in the model. Given the mechanistic model, there is not one effect, but the average or expected reduction in transmission may be able to be found from the model (or at least some summary of the studies that are used to calibrate the model parameters). That would improve interpretation greatly.

Response 2.4

As requested by Reviewer 2 above, we simplified the analysis considering one reference model and a sensitivity analysis (Supplementary Figure S14). This situation also corresponds to the notion of the “true effect” of OM-85 which for the 0% transmission reduction scenario is representative. This “true effect” has been calibrated to the control-group RTI rate dependent (and therefore non-constant) efficacy distribution as meta-analyzed by Yin, including the variability (see Supplementary Figure S9-S11 and Supplementary Methods, Section "Calibration of OM-85 clinical efficacy" for the precise description of how studies from the meta-analysis were selected). In absence of clinical data of OM-85 for non-zero transmission reduction scenarios, no benchmark of true effect of OM-85 in those conditions can be given and we have to rely on the mechanistic nature of our model to translate rate reduction to endpoint change (the immunological effect of the product is assumed to be independent from NPI).

- Fig. 3: In the discussion, the authors cite hypothesis testing as the main value of the RTI rate ratio effect size. Given that and the importance of hypothesis testing in such trials, the empirical power of the trials should be displayed in the main text.

Response 2.5

We thank the reviewer for raising this issue and added empirical power (with sample size of 50 patients per arm) in the main text of the manuscript.

Second paragraph of Section "Effect of NPI scenario on recruitment " of the Results (Line 130-132), now ends with: " Note that empirical power values at fixed sample size of 50 patients per arm also follows this trend (no reduction of the transmission rate: 0.86, 5% reduction: 0.86, 15% reduction: 0.76 and 25% reduction: 0.34)."

- Fig. 3c: I would like to see some discussion of the increased (maximum 75th percentile) RTI rate ratio observed under the strong NPI. Do the authors think this is likely an outlier simulation result, or is there some reason that might occur?

Response 2.6

Reviewer #1 made a very similar comment which we answered in "Response 1.2".

- P. 3, l. 124: is the dispersion coefficient that is used estimated from the no-reduction scenario in all cases or is it varied by transmission reduction scenario? The former is more realistic for trial planning, since investigators are more likely to have detailed information on counts in non-NPI settings than NPI settings. My guess is using the former will lead to more extreme sample size differences than the latter, but it would be interesting to know how the dispersion coefficient changes with increased NPI strength.

Response 2.7

The dispersion coefficient used is estimated for each transmission reduction scenario (taking the average coefficient from the treatment and the placebo arms). We clarified this in the Methods Section (see "Sample size and recruitment estimation"). Indeed, using a constant dispersion coefficient obtained from the no-reduction scenario is more realistic in the context of trial design. We explored whether this would result, as the reviewer suggests, in more extreme sample size

differences (see Figure below, which we added as Supplementary Figure S15, referenced in the Methods, Line 366).

Supplementary Figure S15. Sensitivity of sample size (per arm) to using a dispersion parameter in the calculation either fixed or varied per NPI scenario. Sample size calculations were performed as in Zhu et al. (2013) [10.1002/sim.5947] (see Methods). For this, we conducted generalized linear regression analysis on number of RTIs with negative binomial distributions to get the mean and dispersion parameter for each NPI scenario and for each arm (placebo and treated). We then either used the dispersion parameter of the placebo arm from the scenario without reduction of transmission ("Fixed", hatched bars) or the average dispersion parameter between the two arms per NPI scenario ("Varied", solid bars).

Though indeed, when computed with fixed dispersion coefficient (from no-reduction scenario), the sample sizes are overall slightly higher than when computed with varied (on an NPI-scenario basis) dispersion coefficient. However, this difference remains negligible compared to the sample size's increase when transmission is reduced.

We clarified in the Methods Section (see "Sample size and recruitment estimation") how the dispersion parameters were estimated, Lines 339-342 now reads: "Calculations employed the power.nb.test function of the MKmisc package given the ratio of rates in both trial arms, average dispersion parameter between both arms (thus varied per NPI scenario; note that fixed dispersion parameter result in similar results, Supplementary Figure S15), an alpha of 0.025 and correction for average study duration (e.g. due to dropout $\kappa=0.75$)."

Following the reviewer's suggestion, we then explored how the dispersion coefficient is evolving with increased transmission reduction. As noted by the reviewer, the dispersion coefficient is decreasing with decreasing transmission (or with increasing NPI strength).

- Discussion: the numerical results presented are very specific to the setting under examination. As stated above, identifying some generalizable method for adjusting sample size/effect size would be very valuable. If that is not possible, however, the authors should clearly state this limitation and the key factors that may change the magnitude of the discrepancy in effect sizes under NPIs.

Response 2.8

We are grateful for the reviewer's viewpoint that the conclusion from this study could be enhanced. We agree that an adjustment method for the sample size would be straightforward to suggest from our results (at least an estimation) but we fear that such suggestion could be easily misinterpreted in view of the multitude of factors determining the success / feasibility of a trial and potential mitigation strategies in practice. We therefore would like to point to Response 2.1, which compiles a comprehensive NPI-strength resolved risk mitigation strategy that concludes from our findings of NPI-strength impact on clinical significance, power, and recruitment and which was integrated into the manuscript as Table 1.

Minor Comments:

- P. 2, ll. 37–40: This sentence seems to be cut off in the middle; I also don't understand what is meant by "respiratory disease prophylaxis and international trials". I expect that all trials, whether within one country or multi-site trials across countries, will be affected.

Response 2.9

We replaced "international trials" by "trials across the world". Furthermore, we fixed the problem with the cut-off sentence by elaborating on how execution of clinical trials is different in pandemic trials than what pre-pandemic studies tell us.

Lines 37-40 now reads: "At the same time, respiratory disease prophylaxis and trials across the world are strongly affected by these drastic changes because the design of clinical trials is usually conceived from pre-pandemic settings, e.g. the sample size calculation and the choice of endpoints and eligibility are based on historical interventional and observational studies and do not mirror the current pandemic situation.

- Fig. 2: Some mention should be made of the relatively poor fit of the model in January–March, and whether this raises concern for the usefulness of the results.

Response 2.10

We thank the reviewer for pointing out that we should explain more the suboptimal fit between January and March. This is due to several reasons:

Firstly, the 5-year average of the RGCP data was not used for calibrating the epidemiological model which was calibrated with heterogeneous data on the main respiratory viruses (Supplementary Methods). As a result, the 5-year average of the RGCP data can be seen as a validation for the prior calibration of the virus-specific infection dynamics.

We specified this accordingly in the main text with Lines 64-67 reading: "As representative comparator and validation of the prior virus-specific infection dynamics calibration, we used the 5-year average and the 2019-2020 upper and lower RTI (URTI and LRTI) incidence from the

communicable and respiratory disease report 2019 to 2020 published in the UK by the Royal College of General Practitioners (RCGP) (points and full lines in Figure 2b)."

Second, our calibration data included epidemiologic data on RSV and Influenza, which both had a similar time point of maximal incidence around January to February. Thus, the overall magnitude of RTI incidence in this time frame is increased. Other authors such as Reeves et al. 2017 (DOI: 10.1111/irv.12443) and Paget et al., 2010 (DOI: 10.1007/s00431-010-1164-0) report the same behavior for both RSV and influenza, thus, we believe that the calibration sources that we chose are appropriate in this regard.

Third, Figure S1 in the Supplementary (see below) shows that while the RCGP data do not exhibit a pronounced peak for upper RTIs (URTIs) around January as suggested by our model prediction, they do display one for lower RTIs (LRTIs) in agreement with our model simulations.

We assumed in our model that the ratio of the number of LRTIs and number of RTIs (resp. URTIs to RTIs) stays constant over the whole course of simulation. The figure below shows this ratio as predicted by our model and for the RCGP 5-year average. While the agreement is very good throughout the year, validating our assumption, there is some discrepancy in January (Day 0 in the figure below) which corresponds to the region of suboptimal fit. This simplifying hypothesis (constant ratios of URTIs/LRTIs) may be limiting in specific use cases where a finer representation of distribution of URTIs compared to LRTIs may be needed but do not think it is limiting for the results presented in this manuscript.

Another possibility may be that URTIs, because they are milder compared to LRTIs, especially in children (Feldman et al., 2015, DOI: 10.1089/ped.2014.0463), may be underestimated in

frequency, as the RCGP data are based on the reports from GPs. And if a person has symptoms (cough, fever, pharyngitis...) which, accumulated, would be classified as a URTI, they do not necessarily seek a general practitioner (even before the start of the Covid-19 pandemic). Hence, the URTI burden might be underestimated in the RCGP data.

- P. 3, ll. 79–80: It would be useful here to clarify that the intervention/follow-up period of the trial is only the second year (coincident with the NPIs).

Response 2.11

We followed the reviewer's suggestion and clarified this point. The beginning of Section " Effect of NPIs on efficacy of RTI prophylaxis" of the Results (Lines 80-84) now reads: "To represent the effect of different NPI scenarios for a 2-year clinical trial (where the first year is the selection period and the second year is the intervention period), we defined scenarios where the transmission rate is decreased by 5%, 15% and 25% during the second year (Figure 3a). As a result, the selection year is unaffected by NPIs and the intervention period overlaps with the NPIs."

- Fig. 3f: This is not very useful with the large y-axis. I think this panel would be improved by showing only the 0, 5, and 15% reduction scenarios.

Response 2.12

We followed the reviewer's suggestion and removed 25% reduction scenario from Figure 3f and reduced the scale of the y-axis.

- P. 3, ll. 99–106: I think it would be easier to interpret the other results if the reduction between Year 1 and Year 2 in R_c were introduced first, so I recommend moving this paragraph above the previous two.

Response 2.13

We understand the reviewer's suggestion, however we think this paragraph best belongs after the previous two since it is discussing the results of the Effect Model analysis (Figure 3d) which is harmonizing the results of the other efficacy metrics (Figure 3b-c) and as such we think it is natural if presented after. To improve the clarity of the results discussion in this part, we now define R_c and R_t in the first paragraph (Line 90-91: "We define here the RTI rate as the number of RTIs counted during the 12-month follow-up period (year 2 of the trial): R_t for the treated group and R_c for the control group.>").

- P. 3, ll. 109–114: Can the authors cite guidelines or example trials that use these benchmarks for clinical relevance?

Response 2.14

In the EU referral procedure EMEA/H/A-31/1465¹, CHMP states for studies with up to 0.9 prevented respiratory tract infection episodes per year that differences are "of questionable clinical relevance". We added a citation to that report.

The section now reads:

"We indicate a region matching three conditions (orange area in Figure 3d, lower right quadrant):
i) recurrent RTI with more than 3 RTIs per year (in real-life clinical trials recurrent RTI is often

1

https://www.ema.europa.eu/en/documents/referral/bacterial-lysate-medicines-article-31-referral-chmp-assessment-report_en.pdf

defined as 3 or more RTIs in the previous year and clinical benefit is considered to prevent recurrence of RTI) as well as ii) an absolute benefit of at least 1 RTI per year under which significant clinical benefit becomes less evident to demonstrate for such products [EMA/457345/2019] and iii) a rate reduction of [...]

- P. 3, I. 128: Why is the NPI now taken to begin in year 1? Clarified in discussion, but should be mentioned here.

Response 2.15

We clarified this section which now reads (Lines 135-138): "We estimated the time required to recruit the estimated sample sizes (Figure 3e) if NPIs were started at the beginning of year 1 and by assuming a constant hypothetical screening rate of 1000 patients per year. Year 1 is the selection year during which patients are screened and possibly included in an in silico trial. NPIs introduced during this period could perturb the selection process."

- P. 4, II. 155–157: Can you clarify these sentences a bit? Is this saying that the model assumes that OM-85 only affects probability of infection of the individual and not their probability of onward transmission?

Response 2.16

The between-host model is used as an input to the within-host model to determine the frequency of viral exposure. Thus, the coupling between the within-host and the between-host model is unidirectional in our case. This is an approximation since most likely OM-85 also affects an individual's transmission (although there are no data on this to our knowledge). This approximation holds as long as a low number of patients are treated (compared to the whole population), which is typically the case for a clinical trial setting (i.e an efficacy setting). If we were to put ourselves in a whole population effectiveness setting, then it would be important to implement this bidirectional coupling. Here, it is also interesting to draw a parallel with vaccine trials in which what matters is efficacy of the vaccine to protect a treated individual from getting infected whereas when real-world effectiveness needs to be assessed by health authorities, the transmission reduction brought by the vaccine also needs to be taken into account.

Reviewer #3:

This is a novel study introducing one in silico approach to replace clinical trials of respiratory tract infections taking into consideration what the impact of a pandemic can be on their feasibility. I have some concerns for this submission.

Response 3.1

We regret that reviewer 3 has gotten the impression that the study aims at replacing clinical trials. This is not the case. We suggest this model as a tool to assess the feasibility of a trial depending on the strength of non-pharmaceutical interventions to mitigate Covid-19. Use of model informed drug development to replace clinical data is possible (e.g. extrapolating efficacy from adults to children based on a given model) but requires extensive validation and qualification of models for this purpose.

In the last paragraph of the Introduction, we thus now detail that: "However, to our knowledge, there is currently no available modeling approach that can simulate RTI prophylaxis trials under

Covid-19 pandemic conditions and that could serve to better inform respiratory disease trial design and development decisions."

- The authors need to decide what they want the outcome of their study to be. It is obvious that they are concerned about the feasibility of running a trial for respiratory tract infections (RTI) taking into consideration the restrictions imposed by COVID-19 pandemic. However they introduce the OM-85 immunomodulator making the reader wondering what is the message the authors wish to confer: provide the efficacy of OM-85 to prevent RTIs; cure RTIs or to predict how long it will take to finalize a study.

Response 3.2

We fully agree with the statement that the study is "concerned about the feasibility of running a trial for respiratory tract infections (RTI) taking into consideration the restrictions imposed by Covid-19 pandemic". Choosing OM-85 as an example treatment seemed reasonable for us as this product is representative of a whole class of products that i) fill a gap of RTI prophylaxis in absence of vaccines (e.g., against RSV) or in addition to existing vaccines (e.g, against influenza, see Esposito et al. 2014, 10.1016/j.vaccine.2014.03.055 which shows no counter indication for co-administration of OM-85 and inactivated influenza vaccine), ii) will need to provide new efficacy data in the near future in light of an EU referral procedure (EMEA/H/A-31/1465) and for which plenty of clinical data are available (more than 50 RCTs, Yin et al 2018).

We changed the last paragraph of the Introduction to highlight the example character of OM-85 for a series of products that have been used in a large number of patients. It now reads:

"After matching known viral disease burden seasonality, intra- and inter-patient variability in RTI resolution and efficacy data, we built a mechanistic model and simulated placebo-controlled *in silico* trials in 1-5 year old pediatric patients with recurrent RTIs (RRTI) treated with an immunomodulating bacterial lysate under 4 different hypotheses of NPI intensities and assessed efficacy and benefit metrics as a function of NPI intensity. We chose the example of OM-85, which is a well characterized (Yin *et al.* 2018) member of a series of bacterial lysates containing medicinal products for respiratory conditions that have been used in over 70 million patients but need to soon provide new clinical efficacy data in view of an EU referral procedure [EMEA/H/A-31/1465], we explored clinical interpretation, power, sample size and recruitment considerations as aspects of trial feasibility."

- The study is inconclusive. The authors feed their program with assumptions which are based on the COVID-19 pandemic trying to interpret it with known epidemiology of non-COVID infections.

Response 3.3

The remark of Reviewer 3 about the opportunity to enhance the conclusion is in line with Reviewer 2. We therefore would like to point to Response 2.1, where we present a summary recommendation table, suggesting concrete NPI-strength resolved risk mitigation strategies for respiratory disease prophylaxis clinical trials.

We also added the following sentence at the end of the Introduction:

"From the simulations, we conclude on Covid-19 pandemic related risk mitigation strategies for efficacy confirmation trials concerning this entire class of products and respiratory tract infection prophylaxis trials in general."

Reviewer #4:

Arsene et al propose a model to predict the impact of Covid-19 on the design of clinical trials of drugs against other respiratory viral diseases. Their starting assumption is that lockdown and other NPI during Covid-19 has led to a change in the epidemiological patterns of respiratory viral diseases, leading to less contacts and less infections, and that this needs to be accounted for when evaluating drugs against viral infections. To address this question the authors use a within-host and between-host model that connects the time-varying prevalence of the infection in the population with the individual risk of developing the infection, and they study the impact of different reductions of virus prevalence on different clinical outcome.

Although the model is conceptually interesting, the implications are fairly limited. There is not much surprise to find that drastic reductions in contacts during lockdowns will reduce the risk of infection, and hence will affect the absolute benefit of RTI interventions as well as the enrollment time. Likewise, it is expected that relative benefits will not be much affected.

Response 4.1

We think that by addressing the other reviewer's comments, we could increase the implications of our study substantially.

For expecting "that relative benefits will not be much affected "; we agree that it seems straightforward to point out, but we hope to have shown that – depending on NPI strength – also the power based on ERR endpoints can become an issue. Especially multi-arm trials might suffer from such decline in power. In fact, the count nature of the outcome is known to be limiting in vaccine trials as well and can hinder reasonable statistical analysis if not planned for. We have drastically expanded the conclusions from this study (Table 1, see Response 2.1) in order to highlight the (unexpected) effect of different NPI strengths on different aspects of trials feasibility (also the power of ERR based endpoints) and risk mitigation strategies.

Moreover, the authors rely on a very specific intervention, OM-85, which further narrows the application of their study.

Response 4.2

We have addressed this concern together with one issue of reviewer 2, regarding the choice of the example OM-85. Please refer to Response 3.2.

Overall I find the model conceptually interesting, but with very limited implication for biology or clinical development.

Response 4.3

We have substantially expanded the conclusions from this study (Table 1, see Response 2.1). Please also refer to Response 3.2 for the representativeness of OM-85 for respiratory tract infection prophylaxis by bacterial lysates used in a large number of patients.

I have some more specific comments:

1) the within-host model is very complicated, and it is not clear at all how these parameters were "calibrated". At least some sensitivity analyses would be needed

Response 4.4

The complexity of our within-host model (13 calibrated parameters) is included within the range of complexity observed with published models of influenza infection (Hancioglu et al. 2006, 10.1016/j.jtbi.2006.12.015; Pawelek et al. 2012, 10.1371/journal.pcbi.1002588; Bacam et al. 2006, 10.1128/JVI.01623-05; Dobrovolny et al. 2010, 10.1371/journal.pone.0013811) or RSV infection (Beauchemin et al. 2019, 10.1371/journal.pone.0214708; González-Parra et al. 2018, 10.1371/journal.pone.0192645; Wethington et al. 2019, 10.1098/rsif.2019.0389).

Those parameters were calibrated using covariance matrix adaptation evolution strategy (CMA-ES) algorithm with mean viral load from experimental human RSV (HRSV) infections (Supplementary Figure S2). We clarified this point in the Supplementary Methods.

We are grateful for the reviewer's suggestion of performing a sensitivity analysis for the within-host model which we address in SI in a dedicated section (Section "Sensitivity analysis of the within-host model", Figure S13).

Supplementary Figure S13. Sensitivity analysis (within-host model). a) Global sensitivity indexes (first (orange bars) and second order interactions (grey bars)) for variance-based global sensitivity analysis on the viremia peak (AUC) with respect to variation of the 13 calibrated parameters of the within-host viral infection disease model around their calibrated value ($\pm 25\%$) using a fractional factorial design of $3^7=2187$ samples from a 3-level design and obtained by Monte Carlo simulations. b) Histogram (density) of relative deviation to the mean of viremia peak AUC in percentage.

Percentage of total variance is high (96%) and second order interactions only represent a negligible part. Four parameters come out as equally most influential: lysis rate of infected cells, immune-activation by the virus, lymphocyte development rate and virions production rate. This indicates that calibration should be focused on these parameters if the model is to be adapted to new viremia data (including different viruses for example) and that experimental efforts could be directed at better informing those parameters to increase robustness in such viral infection models. These results where parameters related to how well the virus replicates and how efficient the immune system is at clearing infected cells are found to be most influential and are consistent with what was reported with a model of viral dynamics for influenza [Pawelek et al. 2012].

Notice that the Virtual Population approach is a way to assess sensitivity of the model. Additionally, we have also varied important parameters not varied in the Virtual Population as

part of an uncertainty analysis as described in Section "Mechanistic uncertainty management" of the Supplement.

2) Since the model not virus specific, I would add that it is not clear why parameters would be similar for different viruses. When looking at viruses for which viral kinetic modeling has been done, parameters are fairly different between SARS-CoV-2, Influenza or RSV

Response 4.5

As noted by the reviewer, if the within-host model were to be made virus-specific, parameters would have to be adapted to the considered virus. We made the choice to have a model which would reproduce an average infection in terms of viral dynamics. For this, considering our target population (young pediatric patients), RSV is the natural choice being the most prevalent in that age group². We recognize that this may represent a limitation of our model but in turn reduces its complexity. For future works, we are considering an extension of the model to be more virus-specific.

3) It is unclear how infection acquisition is considered. The authors mention a "detectable RTI" but it is unclear how this is related to the within-host model.

We thank the reviewer for raising this point, indeed this information was missing, and we have clarified this point in the Supplementary Methods. To declare a RTI, we set the threshold to declare an infection clinically detectable to 20% on the proportion of infected cells.

4) If I were in charge in designing such a clinical trial, I think that I would use as a primary or secondary endpoint a factor related to symptom (severity, duration). I would suspect much more statistical power to detect a drug effect on these factors than on the acquisition. This could be an interesting aspect to consider in such a detailed analysis on designing clinical trials in RTI.

We thank the reviewer for this suggestion. We plan for future works to perform a study on such factors which could affect trial design in RTI prophylaxis and we will include symptoms-related endpoints as an alternative to RTI count. For the current study, we chose to restrict ourselves to canonical endpoints in RTI prophylaxis trials but indeed to address the effects of Covid-19 on trials, potentially a different endpoint less affected could be beneficial. However, in the same way as the RTI count, symptoms are expected to be highly affected since they also depend on the viral exposition dynamics.

Furthermore, as noted in Response 4.3, we expanded the conclusions from this study (Table 1, see Response 2.1). In this Table, we indicate measures which could be taken to mitigate risk of trial failure depending on NPI strength, notably for weak NPIs: "Include secondary endpoints that add a diversified and multi-faceted view to the clinical significance for assessors of the trial results (e.g. symptom-free days as RTI-duration related endpoint)".

We hope that we have addressed all outstanding issues and can resolve all comments by the reviewers by our additional explanations, changes to the text, additional analyses and revised figures. We would like to thank all reviewers again for the constructive feedback. Please do not hesitate to get back to us. We are more than happy to explain further and are always open to discussions.

² Zhang, D., He, Z., Xu, L., Zhu, X., Wu, J., Wen, W., Zheng, Y., Deng, Y., Chen, J., Hu, Y., Li, M., & Cao, K. (2014). Epidemiology characteristics of respiratory viruses found in children and adults with respiratory tract infections in southern China. In *International Journal of Infectious Diseases* (Vol. 25, pp. 159–164). Elsevier BV. <https://doi.org/10.1016/j.ijid.2014.02.019>

We look forward to receiving your feedback.

Sincerely,

Alexander Kulesza

On behalf of the authors

NAME | ORCID

Simon Arsene | 0000-0002-8040-8537

Claire Couty | 0000-0001-9167-7428

Igor Faddeenkov | 0000-0001-5735-0173

Natacha Go | 0000-0003-4674-3726

Solene Granjeon-Noriot | 0000-0002-8360-0938

Daniel Smit | 0000-0002-7676-9110

Riad Kahoul | 0000-0002-6181-7466

Ben Illigens | 0000-0003-0683-0809

Jean-Pierre Boissel | 0000-0001-8938-2284

Aude Chevalier | NA

Lorenz Lehr | NA

Christian Pasquali | 0000-0002-2117-8487

Alexander Kulesza | 0000-0002-8812-8548

Reviewers' Comments:

Reviewer #1:

Remarks to the Author:

The authors have sufficiently addressed all of my concerns.

Reviewer #2:

Remarks to the Author:

I thank the authors for their careful consideration of the comments of all of the reviewers, and for their thoughtful and responsive revision. All of my major concerns have been addressed. In particular, I feel that the addition of Table 1 and the clarification of the discussion greatly improve the paper.

I only have one minor comment: Use of the notation R_t for the rate of RRTIs under treatment may be a bit confusing given the common use of R_t for the time-varying reproduction number in an epidemic (i.e., COVID rates) and specifically as the target of NPIs. Another notation might be less confusing.

Reviewer #3:

Remarks to the Author:

None

Reviewer #4:

Remarks to the Author:

I thank the authors for their thorough sensitivity analysis on the within-host model. However I regret to say that this does not change my initial review.

Reading the paper (including the title), the reader is under the misleading impression that the results of this work are really due to the novelty of combining between- and within-host models, but I doubt it is really the case. The authors indeed use an overly complicated within-host model to address their question, but the model remains calibrated on a single viral infection (RSV) and a very specific treatment, bringing their analysis to an unnecessary level of detail. For instance, do we really need to integrate the PB/PK of OM-85 to study the general effect of NPI on RTI trials? In fact in my opinion, the authors failed to show that their within-host model brings any new insight as compared to a simple epidemiological model that would consider that the risk of disease acquisition is reduced by a given factor in individuals treated with OM-85 (with possibly a time-dependent effect taking into account drug wash-out). In order for the within-host model to be really valuable, one would need to integrate more subtle aspects, such as the benefit resulting from an imperfect drug protection on disease severity and transmission.

Consequently I don't think that the complexity of the model really helps better understand the interaction of RTI infection and treatment, nor improves the design of RTI designs. I would recommend to either focus on a simpler model to make a general case on the effects of RTI on study designs, or to publish in a more specialized pharmacology or modeling journal (such as Clinical Pharmacology & Therapeutics, or PLoS Computational Biology) the specific analysis of their compound at the within and between host levels.

Novadiscovery SA

Alexander Kulesza, PhD

Team Leader | Biomodeling and Simulation

1 Place Verrazzano, 69009 Lyon, France

+33 7 82 92 44 620

alexander.kulesza@novadiscovery.com

novadiscovery.com

Lyon, March 10th 2022

Final Manuscript Revision

Dear Reviewers

Thank you for your feedback on our revised manuscript.

We have attempted to address outstanding issues of reviewer 2 and 4 and suggest a few changes in the manuscript are highlighted in red for addition / modification and strike-through for deletion. We additionally provide a clean version.

REVIEWERS' COMMENTS

Reviewer #2 (Remarks to the Author):

I thank the authors for their careful consideration of the comments of all of the reviewers, and for their thoughtful and responsive revision. All of my major concerns have been addressed. In particular, I feel that the addition of Table 1 and the clarification of the discussion greatly improve the paper.

I only have one minor comment: Use of the notation R_t for the rate of RRTIs under treatment may be a bit confusing given the common use of R_t for the time-varying reproduction number in an epidemic (i.e., COVID rates) and specifically as the target of NPIs. Another notation might be less confusing.

Response:

We agree with the reviewer on the confusion which can arise from the use of R_t in such a context, however, we would like to keep this notation which is in line with existing work on the Effect Model methodology (Boissel et al. 1993, DOI:10.1097/00005344-199309000-00003; Wang et al. 2009, DOI:10.1186/1742-7622-6-1; Boissel et al. 2011, DOI:10.2217/pme.11.54) and thus added a footnote at the first mention of R_t which reads:

“ R_t may also refer to the (time-varying) reproduction number of an epidemic but we refer to the rate (or risk) of a specific event, here an RTI, in line with earlier work using the Effect Model methodology.”

Reviewer #4 (Remarks to the Author):

Reading the paper (including the title), the reader is under the misleading impression that the results of this work are really due to the novelty of combining between- and within-host models, but I doubt it is really the case.

Response:

See our response below, notice that the title is also changed and the reviewers' comment on the misinterpretation of the title should be lifted.

The authors indeed use an overly complicated within-host model to address their question, but the model remains calibrated on a single viral infection (RSV) and a very specific treatment, bringing their analysis to an unnecessary level of detail. For instance, do we really need to integrate the PB/PK of OM-85 to study the general effect of NPI on RTI trials ?

Response:

We agree with the reviewer that the model deviates in part from the principle of parsimony, which is due to several facts. First, the approach we follow is rooted in physiology and systems biology, which means that larger scale behaviour can emerge from the composition of individual components' functionality. There are challenges for such a model, such as parameter identifiability for example, nevertheless, complex models for in silico clinical trials emerge also in the field of regulatory decision making (Musuamba et al. 2021 DOI:10.1002/psp4.12669). The validation of the model for a given context of use determines whether it is qualified or not, and the actual complexity of the model is less important for its fitness for purpose (still, this is a field of ongoing research). We have indicated these developments and cited an additional reference to be more transparent of the challenges for such models but also of their promise.

Lines 265-268 (Section "Modeling approach" in the Methods) now reads:

It is to note that the model is rather complex thanks to its roots in systems biology and quantitative systems pharmacology. Despite the deviation from the principle of parsimony for this study, such type of models can be used for impactful (e.g. regulatory) decision making, when properly validated [Musuamba2021].

In fact in my opinion, the authors failed to show that their within-host model brings any new insight as compared to a simple epidemiological model that would consider that the risk of disease acquisition is reduced by a given factor in individuals treated with OM-85 (with possibly a time-dependent effect taking into account drug wash-out).

Response:

The members of the NIH Interagency Modeling and Analysis Group (IMAG), from the working group Multiscale Modeling and Viral Pandemics (MSM) have reviewed the field (Karr et al. 2022 DOI:10.3389/fsysb.2022.822606) noting that "Multi-scale within-host modeling is common, with scales from the molecular and cellular levels integrated successfully with the larger whole-organ or whole-body scales. There are fewer models, however, that successfully combine within-host models with

population-level models. One barrier to the development of such models is the potential to lose the within-host granularity that is often seen when integrating to a higher scale.” See a few references within.

We are therefore convinced that our approach to use randomly sampled instantaneous infection hazard informed by the population scale model and triggering discrete viral exposures for individual patients is novel and advances the progress of this community. We have added the following statement into the introduction.

Lines 48-50 of the Introduction now reads:

As mentioned in a recent review by Karr et al. multi-scale within-host modelling is common, but there are much fewer models that interface within-host models with between-host models because the within-host granularity risks getting lost when integrating to a higher scale \cite{10.3389/fsysb.2022.822606}.

In order for the within-host model to be really valuable, one would need to integrate more subtle aspects, such as the benefit resulting from an imperfect drug protection on disease severity and transmission.

In a clinical trial setting, only very few individuals obtain the investigational drug and thus, feedback from OM-85 treated individuals in that trial on the overall disease burden (especially in multicenter trials) should be negligible. We agree that this model should therefore be applied to a clinical trial setting until this limitation has been lifted.

We hope that we have addressed all outstanding issues and can resolve all comments by the reviewers by our additional explanations, changes to the text, additional analyses and revised figures. We would like to thank all reviewers again for the constructive feedback. Please do not hesitate to get back to us. We are more than happy to explain further and are always open to discussions.

We look forward to receiving your feedback.

Sincerely,

Alexander Kulesza

On behalf of the authors

NAME | ORCID

Simon Arsene | 0000-0002-8040-8537

Claire Couty | 0000-0001-9167-7428

Igor Faddeenkov | 0000-0001-5735-0173

Natacha Go | 0000-0003-4674-3726

Solene Granjeon-Noriot 0000-0002-8360-0938

Daniel Smit | 0000-0002-7676-9110

Riad Kahoul | 0000-0002-6181-7466
Ben Illigens | 0000-0003-0683-0809
Jean-Pierre Boissel | 0000-0001-8938-2284
Aude Chevalier | NA
Lorenz Lehr | NA
Christian Pasquali | 0000-0002-2117-8487
Alexander Kulesza | 0000-0002-8812-8548